# DualDICE: Behavior-Agnostic Estimation of Discounted Stationary Distribution Corrections

Ofir Nachum [* 1]   Yinlam Chow [* 1]   Bo Dai [1]   Lihong Li [1]

## Abstract

In many real-world reinforcement learning applications, access to the environment is limited to a fixed dataset, instead of direct (online) interaction with the environment. When using this data for either evaluation or training of a new policy, accurate estimates of *discounted stationary distribution ratios* — correction terms which quantify the likelihood that the new policy will experience a certain state-action pair normalized by the probability with which the state-action pair appears in the dataset — can improve accuracy and performance. In this work, we propose an algorithm, DualDICE, for estimating these quantities. In contrast to previous approaches, our algorithm is agnostic to knowledge of the behavior policy (or policies) used to generate the dataset. Furthermore, it eschews any direct use of importance weights, thus avoiding potential optimization instabilities endemic of previous methods. In addition to providing theoretical guarantees, we present an empirical study of our algorithm applied to off-policy policy evaluation and find that our algorithm significantly improves accuracy compared to existing techniques.

## 1. Introduction

Reinforcement learning (RL) has recently demonstrated a number of successes in various domains, such as games (Mnih et al., 2013), robotics (Andrychowicz et al., 2018), and conversational systems (Gao et al., 2019; Li et al., 2016). These successes have often hinged on the use of simulators to provide large amounts of experience necessary for RL algorithms. While this is reasonable in game environments, where the game is often a simulator itself, and some simple real-world tasks can be simulated to an accurate enough degree, in general one does not have such direct

or easy access to the environment. Furthermore, in many real-world domains such as medicine (Murphy et al., 2001), recommendation (Li et al., 2011), and education (Mandel et al., 2014), the deployment of a new policy, even just for the sake of performance evaluation, may be expensive and risky. In these applications, access to the environment is usually in the form of *off-policy* data (Sutton & Barto), logged experience collected by potentially multiple and possibly unknown *behavior* policies.

State-of-the-art methods which consider this more realistic setting — either for policy evaluation or policy improvement — often rely on estimating (*discounted*) *stationary distribution ratios* or *corrections*. For each state and action in the environment, these quantities measure the likelihood that one's current *target* policy will experience the state-action pair normalized by the probability with which the state-action pair appears in the off-policy data. Proper estimation of these ratios can improve the accuracy of policy evaluation (Liu et al., 2018) and the stability of policy learning (Gelada & Bellemare, 2018; Hallak & Mannor, 2017; Liu et al., 2019; Sutton et al., 2016). In general, these ratios are difficult to compute, let alone estimate, as they rely not only on the probability that the target policy will take the desired action at the relevant state, but also on the probability that the target policy's interactions with the environment dynamics will lead it to the relevant state.

Several methods to estimate these ratios have been proposed recently (Gelada & Bellemare, 2018; Hallak & Mannor, 2017; Liu et al., 2018), all based on the steady-state property of stationary distributions of Markov processes (Hastings, 1970). This property may be expressed locally with respect to state-action-next-state tuples, and is therefore amenable to stochastic optimization algorithms. However, these methods possess several issues when applied in practice: First, these methods require knowledge of the probability distribution used for each sampled action appearing in the off-policy data. In practice, these probabilities are usually not known and difficult to estimate, especially in the case of multiple, non-Markovian behavior policies. Second, the loss functions of these algorithms involve per-step importance ratios (the ratio of action sample probability with respect to the target policy versus the behavior policy). Depending on how far the behavior policy is from the target policy, these quan-

---

[*]Equal contribution  [1]Google Research. Correspondence to: Ofir Nachum <ofirnachum@google.com>.

*Reinforcement Learning for Real Life (RL4RealLife) Workshop in the 36th International Conference on Machine Learning*, Long Beach, California, USA, 2019. Copyright 2019 by the author(s).

tities may have large variance, and thus have a detrimental effect on stochastic optimization algorithms.

In this work, we propose *Dual stationary DIstribution Correction Estimation (DualDICE)*, a new method for estimating discounted stationary distribution ratios. It is agnostic to the number or type of behavior policies used for collecting the off-policy data. Moreover, the objective function of our algorithm does not involve any per-step importance ratios, and so our solution is less likely to be affected by their high variance. We provide theoretical guarantees on the convergence of our algorithm and evaluate it on a number of off-policy policy evaluation benchmarks. We find that DualDICE can consistently, and often significantly, improve performance compared to previous algorithms for estimating stationary distribution ratios.

## 2. Background

We consider a Markov Decision Process (MDP) setting (Puterman, 1994), in which the environment is specified by a tuple $\mathcal{M} = \langle S, A, R, T, \beta \rangle$, consisting of a state space, an action space, a reward function, a transition probability function, and an initial state distribution. A policy $\pi$ interacts with the environment iteratively, starting with an initial state $s_0 \sim \beta$. At step $t = 0, 1, \cdots$, the policy produces a distribution $\pi(\cdot|s_t)$ over the actions $A$, from which an action $a_t$ is sampled and applied to the environment. The environment stochastically produces a scalar reward $r_t \sim R(s_t, a_t)$ and a next state $s_{t+1} \sim T(s_t, a_t)$. In this work, we consider infinite-horizon environments and the $\gamma$-discounted reward criterion for $\gamma \in [0, 1)$. It is clear that any finite-horizon environment may be interpreted as infinite-horizon by considering an augmented state space with an extra terminal state which continually loops onto itself with zero reward.

### 2.1. Off-Policy Policy Evaluation

Given a *target* policy $\pi$, we are interested in estimating its value, defined as the normalized expected per-step reward obtained by following the policy:

$$\rho(\pi) := (1 - \gamma) \cdot \mathbb{E}\big[ \textstyle\sum_{t=0}^{\infty} \gamma^t r_t \mid s_0 \sim \beta, \forall t,$$
$$a_t \sim \pi(s_t), r_t \sim R(s_t, a_t), s_{t+1} \sim T(s_t, a_t)\big]. \quad (1)$$

The off-policy policy evaluation (OPE) problem studied here is to estimate $\rho(\pi)$ using a fixed set $\mathcal{D}$ of transitions $(s, a, r, s')$ sampled in a certain way. This is a very general scenario: $\mathcal{D}$ can be collected by a single behavior policy (as in most previous work), multiple behavior policies, or an oracle sampler, among others. In the special case where $\mathcal{D}$ contains entire trajectories collected by a known behavior policy $\mu$, one may use *importance sampling* (IS) to estimate $\rho(\pi)$. Specifically, given a finite-length trajectory $\tau = (s_0, a_0, r_0, \ldots, s_H)$ collected by $\mu$, the IS estimate of

$\rho$ based on $\tau$ is estimated by (Precup, 2000):

$$(1 - \gamma) \left( \prod_{t=0}^{H-1} \frac{\pi(a_t|s_t)}{\mu(a_t|s_t)} \right) \left( \sum_{t=0}^{H-1} \gamma^t r_t \right).$$

Although many improvements exist (e.g., Farajtabar et al., 2018; Jiang & Li, 2016; Precup, 2000; Thomas & Brunskill, 2016), importance-weighting the entire trajectory can suffer from exponentially high variance, which is known as "the curse of horizon" (Li et al., 2015; Liu et al., 2018).

To avoid exponential dependence on trajectory length, one may weight the states by their *long-term* occupancy measure. First, observe that the policy value may be re-expressed as,

$$\rho(\pi) = \mathbb{E}_{(s,a) \sim d^\pi, r \sim R(s,a)}[r],$$

where

$$d^\pi(s, a) := (1 - \gamma) \textstyle\sum_{t=0}^{\infty} \gamma^t \Pr\big(s_t = s, a_t = a$$
$$\mid s_0 \sim \beta, \forall t, a_t \sim \pi(s_t), s_{t+1} \sim T(s_t, a_t)\big), \quad (2)$$

is the *normalized discounted stationary distribution* over state-actions with respect to $\pi$. One may define the discounted stationary distribution over states analogously, and we slightly abuse notation by denoting it as $d^\pi(s)$; note that $d^\pi(s, a) = d^\pi(s)\pi(a|s)$. If $\mathcal{D}$ consists of trajectories collected by a behavior policy $\mu$, then the policy value may be estimated as,

$$\rho(\pi) = \mathbb{E}_{(s,a) \sim d^\mu, r \sim R(s,a)} \big[ w_{\pi/\mu}(s, a) \cdot r \big],$$

where $w_{\pi/\mu}(s, a) = d^\pi(s, a)/d^\mu(s, a)$ is the *discounted stationary distribution correction*. The key challenge is in estimating these correction terms using data drawn from $d^\mu$.

### 2.2. Learning Stationary Distribution Corrections

We provide a brief summary of previous methods for estimating the stationary distribution corrections. The ones that are most relevant to our work are a suite of recent techniques (Gelada & Bellemare, 2018; Hallak & Mannor, 2017; Liu et al., 2018), which are all essentially based on the following steady-state property of stationary Markov processes:

$$d^\pi(s') = (1 - \gamma)\beta(s') +$$
$$\gamma \textstyle\sum_{s \in S} \sum_{a \in A} d^\pi(s)\pi(a|s)T(s'|s, a), \quad \forall s' \in S, \quad (3)$$

where we have simplified the identity by restricting to discrete state and action spaces. This identity simply reflects the conservation of flow of the stationary distribution: At each timestep, the flow out of $s'$ (the LHS) must equal the flow into $s'$ (the RHS). Given a behavior policy $\mu$, equation 3 can be equivalently rewritten in terms of the stationary distribution corrections, i.e., for any given $s' \in S$,

$$\mathbb{E}_{(s_t, a_t, s_{t+1}) \sim d^\mu} \big[ \text{TD}(s_t, a_t, s_{t+1} \mid w_{\pi/\mu}) \mid s_{t+1} = s' \big] = 0, \quad (4)$$

where the following quantity can be viewed as a *temporal difference* associated with $w_{\pi/\mu}$:

$$\text{TD}(s, a, s' \mid w_{\pi/\mu}) := -w_{\pi/\mu}(s') + \gamma w_{\pi/\mu}(s) \cdot \frac{\pi(a|s)}{\mu(a|s)}$$
$$+ (1-\gamma)\beta(s'),$$

provided that $\mu(a|s) > 0$ whenever $\pi(a|s) > 0$. Accordingly, previous works optimize loss functions which minimize this TD error using samples from $d^\mu$. We emphasize that although $w_{\pi/\mu}$ is associated with a temporal difference, it does not satisfy a Bellman recurrence in the usual sense (Bellman, 2003). Indeed, note that equation 3 is written "backwards": The occupancy measure of a state $s'$ is written as a (discounted) function of *previous* states, as opposed to vice-versa. This will serve as a key differentiator between our algorithm and these previous methods.

### 2.3. Off-Policy Estimation with Multiple Unknown Behavior Policies

While the previous algorithms are promising, they have several limitations when applied in practice:

- The off-policy experience distribution $d^\mu$ is with respect to a single, Markovian behavior policy $\mu$, and this policy must be known during optimization. In practice, off-policy data often comes from multiple, unknown behavior policies.
- Computing the TD error in equation 4 requires the use of per-step importance ratios $\pi(a_t|s_t)/\mu(a_t|s_t)$ at every state-action sample $(s_t, a_t)$. Depending on how far the behavior policy is from the target policy, these quantities may have high variance, which can have a detrimental effect on the convergence of any stochastic optimization algorithm that is used to estimate $w_{\pi/\mu}$.

The method we derive below will be free of the aforementioned issues, avoiding unnecessary requirements on the form of the off-policy data collection as well as explicit uses of importance ratios. Rather, we consider the general setting where $\mathcal{D}$ consists of *transitions* sampled in an unknown fashion. Since $\mathcal{D}$ contains rewards and next states, we will often slightly abuse notation and write not only $(s, a) \sim d^\mathcal{D}$ but also $(s, a, r) \sim d^\mathcal{D}$ and $(s, a, s') \sim d^\mathcal{D}$, where the notation $d^\mathcal{D}$ emphasizes that, unlike previously, $\mathcal{D}$ is not the result of a single, known behavior policy. The target policy's value can be equivalently written as,

$$\rho(\pi) = \mathbb{E}_{(s,a,r)\sim d^\mathcal{D}} \left[ w_{\pi/\mathcal{D}}(s, a) \cdot r \right], \qquad (5)$$

where the correction terms are given by $w_{\pi/\mathcal{D}}(s, a) := d^\pi(s, a)/d^\mathcal{D}(s, a)$, and our algorithm will focus on estimating these correction terms. Rather than relying on the assumption that $\mathcal{D}$ is the result of a single, known behavior policy, we instead make the following regularity assumption:

**Assumption 1** (Reference distribution property)**.** *For any $(s, a)$, $d^\pi(s, a) > 0$ implies $d^\mathcal{D}(s, a) > 0$. Furthermore, the correction terms are bounded by some finite constant $C$:* $\left\| w_{\pi/\mathcal{D}} \right\|_\infty \leq C.$

## 3. DualDICE

We now develop our algorithm, DualDICE, for estimating the discounted stationary distribution corrections $w_{\pi/\mathcal{D}}(s, a) = d^\pi(s, a)/d^\mathcal{D}(s, a)$. In the OPE setting, one does not have explicit knowledge of the distribution $d^\mathcal{D}$, but rather only access to samples $\mathcal{D} = \{(s, a, r, s')\} \sim d^\mathcal{D}$. Similar to the TD methods described above, we also assume access to samples from the initial state distribution $\beta$. We begin by introducing a key result, which we will later derive and use as the crux for our algorithm.

### 3.1. The Key Idea

Consider the following optimization problem of a (bounded) function $\nu : S \times A \to \mathbb{R}$:

$$\min_{\nu:S\times A\to\mathbb{R}} J(\nu), \qquad (6)$$

with respect to the objective function

$$J(\nu) := \frac{1}{2}\mathbb{E}_{(s,a)\sim d^\mathcal{D}} \left[ (\nu - \mathcal{B}^\pi \nu)(s, a)^2 \right]$$
$$- (1-\gamma) \mathbb{E}_{s_0\sim\beta, a_0\sim\pi(s_0)} \left[ \nu(s_0, a_0) \right].$$

In this formulation we use $\mathcal{B}^\pi$ to denote the expected Bellman operator with respect to policy $\pi$ and zero reward: $\mathcal{B}^\pi\nu(s, a) = \gamma\mathbb{E}_{s'\sim T(s,a), a'\sim\pi(s')}[\nu(s', a')]$. The first term in equation 6 is the expected squared Bellman error with zero reward. This term alone would lead to a trivial solution $\nu^* \equiv 0$, which can be avoided by the second term that encourages $\nu^* > 0$. Together, these two terms result in an optimal $\nu^*$ that places some non-zero amount of Bellman residual at state-action pairs sampled from $d^\mathcal{D}$.

Perhaps surprisingly, as we will show, the Bellman residuals of $\nu^*$ are exactly the desired distribution corrections:

$$(\nu^* - \mathcal{B}^\pi\nu^*)(s, a) = w_{\pi/\mathcal{D}}(s, a). \qquad (7)$$

This key result provides the foundation for our algorithm, since it provides us with a simple objective (relying only on samples from $d^\mathcal{D}$, $\beta$, $\pi$) which we may optimize in order to obtain estimates of the distribution corrections. In the text below, we will show how we arrive at this result. We provide one additional step which allows us to efficiently learn a parameterized $\nu$ with respect to equation 6. We then generalize our results to a family of similar algorithms and lastly present theoretical guarantees.

### 3.2. Derivation

**A Technical Observation**   We begin our derivation of the algorithm for estimating $w_{\pi/\mathcal{D}}$ by presenting the following simple technical observation: For arbitrary scalars

$m \in \mathbb{R}_{>0}, n \in \mathbb{R}_{\geq 0}$, the optimizer of the convex problem $\min_x J(x) := \frac{1}{2}mx^2 - nx$ is unique and given by $x^* = \frac{n}{m}$. Using this observation, and letting $\mathcal{C}$ be some bounded subset of $\mathbb{R}$ which contains $[0, C]$, one immediately sees that the optimizer of the following problem,

$$\min_{x:S \times A \to \mathcal{C}} J_1(x), \qquad (8)$$

whose objective function is defined as

$$J_1(x) := \frac{1}{2}\mathbb{E}_{(s,a)\sim d^{\mathcal{D}}}\left[x(s,a)^2\right] - \mathbb{E}_{(s,a)\sim d^{\pi}}\left[x(s,a)\right],$$

is given by $x^*(s,a) = w_{\pi/\mathcal{D}}(s,a)$ for any $(s,a) \in S \times A$. This result provides us with an objective that shares the same basic form as equation 6. The main distinction is that the second term relies on an expectation over $d^{\pi}$, which we do not have access to.

**Change of Variables** In order to transform the second expectation in equation 8 to be over the initial state distribution $\beta$, we perform the following change of variables: Let $\nu : S \times A \to \mathbb{R}$ be an arbitrary state-action value function that satisfies,

$$\nu(s,a) := x(s,a) + \gamma\mathbb{E}_{s'\sim T(s,a),a'\sim\pi(s')}[\nu(s',a')], \\ \forall (s,a) \in S \times A. \qquad (9)$$

Since $x(s,a) \in \mathcal{C}$ is bounded and $\gamma < 1$, the variable $\nu(s,a)$ is well-defined and bounded. By applying this change of variables, the objective function in 8 can be re-written in terms of $\nu$, and this yields our previously presented objective from equation 6. Indeed, by defining

$$\beta_t(s) := \Pr\left(s = s_t \mid s_0 \sim \beta, a_k \sim \pi(s_k),\right. \\ \left. s_{k+1} \sim T(s_k, a_k), \text{ for } 0 \leq k < t\right),$$

to be the state visitation probability at step $t$ when following $\pi$, immediately the initial condition is equal to $\beta_0 = \beta$. Furthermore, the following chain of equalities holds:

$$\begin{aligned} &\mathbb{E}_{(s,a)\sim d^{\pi}}\left[x(s,a)\right] \\ =\ &\mathbb{E}_{(s,a)\sim d^{\pi}}\left[\nu(s,a) - \gamma\mathbb{E}_{s'\sim T(s,a),a'\sim\pi(s')}[\nu(s',a')]\right] \\ =\ &(1-\gamma)\sum_{t=0}^{\infty}\gamma^t\mathbb{E}_{s\sim\beta_t,a\sim\pi(s)}\big[\nu(s,a) \\ &\qquad\qquad - \gamma\mathbb{E}_{s'\sim T(s,a),a'\sim\pi(s')}[\nu(s',a')]\big] \\ =\ &(1-\gamma)\sum_{t=0}^{\infty}\gamma^t\mathbb{E}_{s\sim\beta_t,a\sim\pi(s)}\left[\nu(s,a)\right] \\ &\qquad -(1-\gamma)\sum_{t=0}^{\infty}\gamma^{t+1}\mathbb{E}_{s\sim\beta_{t+1},a\sim\pi(s)}\left[\nu(s,a)\right] \\ =\ &(1-\gamma)\mathbb{E}_{s\sim\beta,a\sim\pi(s)}\left[\nu(s,a)\right]. \end{aligned}$$

This implies that the Bellman residuals of the optimum of this objective give the desired off-policy corrections:

$$(\nu^* - \mathcal{B}^{\pi}\nu^*)(s,a) = x^*(s,a) = w_{\pi/\mathcal{D}}(s,a). \qquad (10)$$

Equation 6 provides a promising approach for estimating the stationary distribution corrections, since the first expectation is over state-action pairs sampled from $d^{\mathcal{D}}$, while the second expectation is over $\beta$ and actions sampled from $\pi$, both of which we have access to. Therefore, in principle we may solve this problem with respect to a parameterized value function $\nu$, and then use the optimized $\nu^*$ to deduce the corrections. In practice, however, the objective in its current form presents two difficulties:

- The quantity $(\nu - \mathcal{B}^{\pi}\nu)(s,a)^2$ involves a conditional expectation inside a square. In general, when environment dynamics are stochastic and the action space may be large or continuous, this quantity may not be readily optimized using standard stochastic techniques. (For example, when the environment is stochastic, its Monte-Carlo sample gradient is generally biased.)
- Even if one has computed the optimal value $\nu^*$, the corrections $(\nu^* - \mathcal{B}^{\pi}\nu^*)(s,a)$, due to the same argument as above, may not be easily computed, especially when the environment is stochastic or the action space continuous.

**Exploiting Fenchel Duality** We solve both difficulties listed above in one step by exploiting Fenchel duality (Rockafellar, 2015): Any convex function $f(x)$ may be written as $f(x) = \max_{\zeta} x \cdot \zeta - f^*(\zeta)$, where $f^*$ is the Fenchel conjugate of $f$. In the case of $f(x) = \frac{1}{2}x^2$, the Fenchel conjugate is given by $f^*(\zeta) = \frac{1}{2}\zeta^2$. Thus, we may express our objective as,

$$\min_{\nu:S \times A \to \mathbb{R}} J(\nu) := \mathbb{E}_{(s,a)\sim d^{\mathcal{D}}}\left[\max_{\zeta}\left(\nu - \mathcal{B}^{\pi}\nu\right)(s,a)\cdot\zeta - \frac{1}{2}\zeta^2\right] \\ - (1-\gamma)\,\mathbb{E}_{s_0\sim\beta,a_0\sim\pi(s_0)}\left[\nu(s_0,a_0)\right].$$

By the interchangeability principle (Dai et al., 2016; Rockafellar & Wets, 2009; Shapiro et al., 2009), we may replace the inner max over scalar $\zeta$ to a max over functions $\zeta : S \times A \to \mathbb{R}$ and obtain a min-max saddle-point optimization:

$$\min_{\nu:S \times A \to \mathbb{R}} \max_{\zeta:S \times A \to \mathbb{R}} J(\nu, \zeta), \qquad (11)$$

where the objective function of this problem is given by

$$J(\nu, \zeta) := \mathbb{E}_{(s,a,s')\sim d^{\mathcal{D}},a'\sim\pi(s')}\Big[(\nu(s,a) - \gamma\nu(s',a'))\zeta(s,a) \\ - \zeta(s,a)^2/2\Big] - (1-\gamma)\,\mathbb{E}_{s_0\sim\beta,a_0\sim\pi(s_0)}\left[\nu(s_0,a_0)\right].$$

Using the KKT condition of the inner optimization problem (which is convex and quadratic in $\zeta$), for any $\nu$ the optimal value $\zeta^*_{\nu}$ is equal to the Bellman residual, $\nu - \mathcal{B}^{\pi}\nu$. Therefore, the desired stationary distribution correction can

then be found from the saddle-point solution $(\nu^*, \zeta^*)$ of the minimax problem in equation 11 as follows:

$$\zeta^*(s,a) = (\nu^* - \mathcal{B}^\pi \nu^*)(s,a) = w_{\pi/\mathcal{D}}(s,a). \quad (12)$$

Now we finally have an objective which is well-suited for practical computation. First, unbiased estimates of both the objective and its gradients are easy to compute using stochastic samples from $d^{\mathcal{D}}$, $\beta$, and $\pi$, all of which we have access to. Secondly, notice that the min-max objective function in equation 11 is linear in $\nu$ and concave in $\zeta$. Therefore in certain settings, one can provide guarantees on the convergence of optimization algorithms applied to this objective (see Section 3.4). Thirdly, the optimizer of the objective in equation 11 immediately gives us the desired stationary distribution corrections through the values of $\zeta^*(s,a)$, with no additional computation.

### 3.3. Extension to General Convex Functions

Besides a quadratic penalty function, one may extend the above derivations to a more general class of convex penalty functions. Consider a generic convex penalty function $f : \mathbb{R} \to \mathbb{R}$. Recall that $\mathcal{C}$ is a bounded subset of $\mathbb{R}$ which contains the interval $[0, C]$ of stationary distribution corrections. If $\mathcal{C}$ is contained in the range of $f'$, then the optimizer of the convex problem, $\min_x J(x) := m \cdot f(x) - n$ for $n/m \in \mathcal{C}$, satisfies the following KKT condition: $f'(x^*) = n/m$. Analogously, the optimizer $x^*$ of,

$$\min_{x : S \times A \to \mathcal{C}} J_1(x), \quad (13)$$

whose objective function is denoted by

$$J_1(x) := \mathbb{E}_{(s,a) \sim d^{\mathcal{D}}} \left[ f(x(s,a)) \right] - \mathbb{E}_{(s,a) \sim d^\pi} \left[ x(s,a) \right],$$

satisfies the equality $f'(x^*(s,a)) = w_{\pi/\mathcal{D}}(s,a)$.

With change of variables $\nu := x + \mathcal{B}^\pi \nu$, the above problem becomes,

$$\min_{\nu : S \times A \to \mathbb{R}} J(\nu), \quad (14)$$

where the objective function is given by

$$J(\nu) := \mathbb{E}_{(s,a) \sim d^{\mathcal{D}}} \left[ f((\nu - \mathcal{B}^\pi \nu)(s,a)) \right] - (1-\gamma) \mathbb{E}_{s_0 \sim \beta, a_0 \sim \pi(s_0)} \left[ \nu(s_0, a_0) \right].$$

Applying Fenchel duality to $f$ in this objective further leads to the following saddle-point problem:

$$\min_{\nu : S \times A \to \mathbb{R}} \max_{\zeta : S \times A \to \mathbb{R}} J(\nu, \zeta), \quad (15)$$

where the objective function of this problem is

$$J(\nu, \zeta) := \mathbb{E}_{(s,a,s') \sim d^{\mathcal{D}}, a' \sim \pi(s')} \left[ (\nu(s,a) - \gamma \nu(s',a')) \zeta(s,a) - f^*(\zeta(s,a)) \right] - (1-\gamma) \mathbb{E}_{s_0 \sim \beta, a_0 \sim \pi(s_0)} \left[ \nu(s_0, a_0) \right].$$

By the KKT condition of the inner optimization problem, for any $\nu$ the optimizer $\zeta_\nu^*$ satisfies,

$$f^{*\prime}(\zeta_\nu^*(s,a)) = (\nu - \mathcal{B}^\pi \nu)(s,a). \quad (16)$$

Therefore, using the fact that the derivative of a convex function $f'$ is the inverse function of the derivative of its Fenchel conjugate $f^{*\prime}$, our desired stationary distribution corrections are found by computing the saddle-point $(\zeta^*, \nu^*)$ of the above problem:

$$\begin{aligned}\zeta^*(s,a) &= f'((\nu^* - \mathcal{B}^\pi \nu^*)(s,a)) \\ &= f'(x^*(s,a)) = w_{\pi/\mathcal{D}}(s,a).\end{aligned} \quad (17)$$

Amazingly, despite the generalization beyond the quadratic penalty function $f(x) = \frac{1}{2}x^2$, the optimization problem in equation 15 retains all the computational benefits that make this method very practical for learning $w_{\pi/\mathcal{D}}(s,a)$: All quantities and their gradients may be unbiasedly estimated from stochastic samples; the objective is linear in $\nu$ and concave in $\zeta$, thus is well-behaved; and the optimizer of this problem immediately provides the desired stationary distribution corrections through the values of $\zeta^*(s,a)$, without any additional computation.

This generalized derivation also provides insight into the initial technical result: It is now clear that the objective in equation 13 is the negative Fenchel dual (variational) form of the Ali-Silvey or $f$-divergence, which has been used in previous work to estimate divergence and data likelihood ratios (Nguyen et al., 2010). Despite their similar formulations, we emphasize that the aforementioned dual form of the $f$-divergence is not immediately applicable to estimation of off-policy corrections in the context of RL, due to the fact that samples from distribution $d^\pi$ are unobserved. Indeed, our derivations hinged on two additional key steps: (1) the change of variables from $x$ to $\nu := x + \mathcal{B}^\pi \nu$; and (2) the second application of duality to introduce $\zeta$. Due to these repeated applications of duality in our derivations, we term our method *Dual stationary DIstribution Correction Estimation* (*DualDICE*).

### 3.4. Theoretical Guarantees

In this section, we consider the theoretical properties of DualDICE in the setting where we have a dataset formed by empirical samples $\{s_i, a_i, r_i, s_i'\}_{i=1}^N \sim d^{\mathcal{D}}$, $\{s_0^i\}_{i=1}^N \sim \beta$, and target actions $a_i' \sim \pi(s_i'), a_0^i \sim \pi(s_0^i)$ for $i = 1, \ldots, N$.[1] We will use the shorthand notation $\hat{\mathbb{E}}_{d^{\mathcal{D}}}$ to denote an average over these empirical samples. Although the proposed estimator can adopt general $f$, for simplicity of exposition we restrict to $f(x) = \frac{1}{2}x^2$. We

---

[1]For the sake of simplicity, we consider the batch learning setting with *i.i.d.* samples as in (Sutton et al., 2008). The results can be easily generalized to single sample path with dependent samples (see Appendix).

consider using an algorithm $OPT$ (*e.g.*, stochastic gradient descent/ascent) to find optimal $\nu, \zeta$ of equation 15 within some parameterization families $\mathcal{F}, \mathcal{H}$, respectively. We denote by $\hat{\nu}, \hat{\zeta}$ the outputs of $OPT$. We have the following guarantee on the quality of $\hat{\nu}, \hat{\zeta}$ with respect to the off-policy policy estimation (OPE) problem.

**Theorem 2.** *(Informal) Under some mild assumptions, the mean squared error (MSE) associated with using $\hat{\nu}, \hat{\zeta}$ for OPE can be bounded as,*

$$
\mathbb{E}\left[\left(\hat{\mathbb{E}}_{d^{\mathcal{D}}}\left[\hat{\zeta}\left(s,a\right)\cdot r\right]-\rho(\pi)\right)^2\right]
$$
$$
=\widetilde{\mathcal{O}}\left(\epsilon_{approx}\left(\mathcal{F},\mathcal{H}\right)+\epsilon_{opt}+\frac{1}{\sqrt{N}}\right), \quad (18)
$$

*where the outer expectation is with respect to the randomness of the empirical samples and $OPT$, $\epsilon_{opt}$ denotes the optimization error, and $\epsilon_{approx}\left(\mathcal{F},\mathcal{H}\right)$ denotes the approximation error due to $\mathcal{F}, \mathcal{H}$.*

The sources of estimation error are explicit in Theorem 2. As the number of samples $N$ increases, the statistical error $N^{-1/2}$ approaches zero. Meanwhile, there is an implicit trade-off in $\epsilon_{approx}\left(\mathcal{F},\mathcal{H}\right)$ and $\epsilon_{opt}$. With flexible function spaces $\mathcal{F}$ and $\mathcal{H}$ (such as the space of neural networks), the approximation error can be further decreased; however, optimization will be complicated and it is difficult to characterize $\epsilon_{opt}$. On the other hand, with linear parameterization of $(\nu, \zeta)$, under some mild conditions, after $T$ iterations we achieve provably fast rate, $\mathcal{O}\left(\exp\left(-T\right)\right)$ for $OPT = $ SVRG and $\mathcal{O}\left(\frac{1}{T}\right)$ for $OPT = $ SGD, at the cost of potentially increased approximation error. See the Appendix for the precise theoretical results, proofs, and further discussions.

## 4. Related Work

**Density Ratio Estimation**   Density ratio estimation is an important tool for many machine learning and statistics problems. Other than the naive approach, (i.e., the density ratio is calculated via estimating the densities in the numerator and denominator separately, which may magnify the estimation error), various direct ratio estimators have been proposed (Sugiyama et al., 2012), including the moment matching approach (Gretton et al., 2009), probabilistic classification approach (Bickel et al., 2007; Cheng et al., 2004; Qin, 1998), and ratio matching approach (Kanamori et al., 2009; Nguyen et al., 2010; Sugiyama et al., 2008)

The proposed DualDICE algorithm, as a direct approach for density ratio estimation, bears some similarities to ratio matching (Nguyen et al., 2010), which is also derived by exploiting the Fenchel dual representation of the $f$-divergences. However, compared to the existing direct estimators, the major difference lies in the requirement of the samples from the stationary distribution. Specifically, the existing estimators require access to samples from both

$d^{\mathcal{D}}$ and $d^{\pi}$, which is impractical in the off-policy learning setting. Therefore, DualDICE is uniquely applicable to the more difficult RL setting.

**Off-policy Policy Evaluation**   The problem of off-policy policy evaluation has been heavily studied in contextual bandits (Dudík et al., 2011; Swaminathan et al., 2017; Wang et al., 2017) and in the more general RL setting (Fonteneau et al., 2013; Jiang & Li, 2016; Li et al., 2015; Mahmood et al., 2014; Paduraru, 2013; Precup et al., 2001; 2000; Thomas & Brunskill, 2016; Thomas et al., 2015). Several representative approaches can be identified in the literature. The Direct Method (DM) learns a model of the system and then uses it to estimate the performance of the evaluation policy. This approach often has low variance but its bias depends on how well the selected function class can express the environment dynamics. Importance sampling (IS) (Precup, 2000) uses importance weights to correct the mismatch between the distributions of the system trajectory induced by the target and behavior policies. Its variance can be unbounded when there is a big difference between the distributions of the evaluation and behavior policies, and grows exponentially with the horizon of the RL problem. Doubly Robust (DR) is a combination of DM and IS, and can achieve the low variance of DM and no (or low) bias of IS. Other than DM, all the methods described above require knowledge of the policy density ratio, and thus the behavior policy. Our proposed algorithm avoids this necessity.

## 5. Experiments

We evaluate our method applied to off-policy policy evaluation (OPE). We focus on this setting because it is a direct application of stationary distribution correction estimation, without many additional tunable parameters, and it has been previously used as a test-bed for similar techniques (Liu et al., 2018). In each experiment, we use a behavior policy $\mu$ to collect some number of trajectories, each for some number of steps. This data is used to estimate the stationary distribution corrections, which are then used to estimate the average step reward, with respect to a target policy $\pi$. We focus our comparisons here to a TD-based approach (Gelada & Bellemare, 2018) and weighted step-wise IS (as described in (Liu et al., 2018)), which we and others have generally found to work best relative to common IS variants (Mandel et al., 2014; Precup, 2000). See the Appendix for additional results and implementation details.

We begin in a controlled setting with an evaluation agnostic to optimization issues, where we find that, absent these issues, our method is competitive with TD-based approaches (Figure 1). However, as we move to more difficult settings with complex environment dynamics, the performance of TD methods degrades dramatically, while our method is still able to provide accurate estimates (Figure 2). Finally, we provide an analysis of the optimization behavior of our

method on a simple control task across different choices of function $f$ (Figure 4). Interestingly, although the choice of $f(x) = \frac{1}{2}x^2$ is most natural, we find the empirically best performing choice to be $f(x) = \frac{2}{3}|x|^{3/2}$. All results are summarized for 20 random seeds, with median plotted and error bars at $25^{\text{th}}$ and $75^{\text{th}}$ percentiles.

## 5.1. Estimation Without Function Approximation

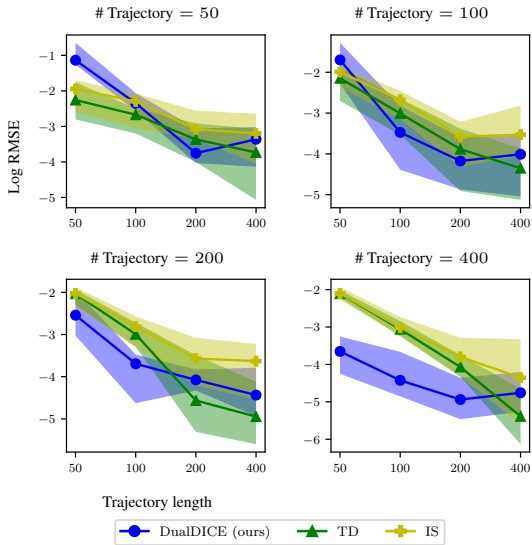

*Figure 1.* We perform OPE on the Taxi domain (Dietterich, 2000). The plots show log RMSE of the estimator across different numbers of trajectories and different trajectory lengths ($x$-axis). For this domain, we avoid any potential issues in optimization by solving for the optimum of the objectives exactly using standard matrix operations. Thus, we are able to see that our method and the TD method are competitive with each other.

We begin with a tabular task, the Taxi domain (Dietterich, 2000). In this task, we evaluate our method in a manner agnostic to optimization difficulties: The objective 6 is a quadratic equation in $\nu$, and thus may be solved by matrix operations. The Bellman residuals (equation 7) may then be estimated via an empirical average of the transitions appearing in the off-policy data. In a similar manner, TD methods for estimating the correction terms may also be solved using matrix operations (Liu et al., 2018). In this controlled setting, we find that, as expected, TD methods can perform well (Figure 1), and our method achieves competitive performance. As we will see in the following results, the good performance of TD methods quickly deteriorates as one moves to more complex settings, while our method is able to maintain good performance, even when using function approximation and stochastic optimization.

## 5.2. Control Tasks

We now move on to difficult control tasks: A discrete-control task Cartpole and a continuous-control task Reacher (Brockman et al., 2016). In these tasks, obser-

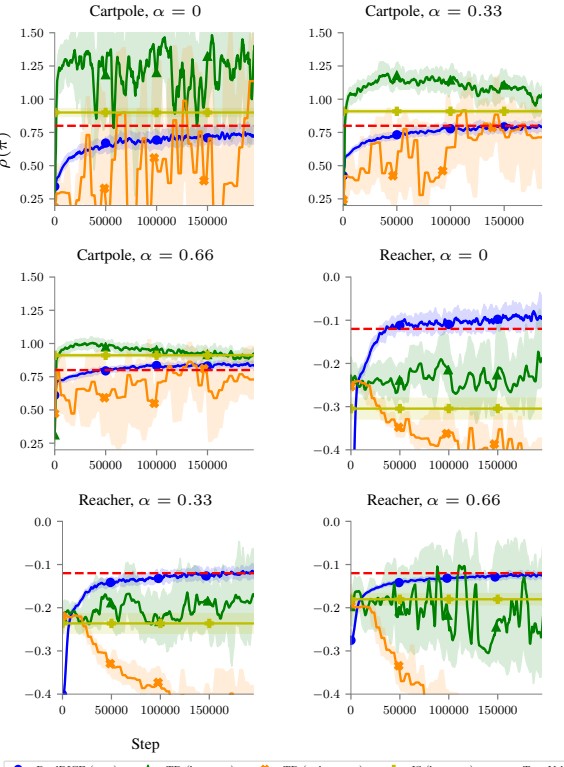

*Figure 2.* We perform OPE on control tasks. Each plot shows the estimated average step reward over training and different behavior policies (higher $\alpha$ corresponds to a behavior policy closer to the target policy). We find that in all cases, our method is able to approximate these desired values well, with accuracy improving with a larger $\alpha$. On the other hand, the TD method performs poorly, even more so when the behavior policy $\mu$ is unknown and must be estimated. While on Cartpole it can start to approach the desired value for large $\alpha$, on the more complicated Reacher task (which involves continuous actions) its learning is too unstable to learn anything at all.

vations are continuous, and thus we use neural network function approximators with stochastic optimization. Figure 2 and Figure 3 show the results of our method compared to the TD method and other common OPE baselines. We find that in this setting, DualDICE is able to provide good, stable performance, while the TD approach suffers from high variance, and this issue is exacerbated when we attempt to estimate $\mu$ rather than assume it as given. On the other hand, in the case when the behavior policy is unknown, empirical results of most other baseline OPE methods either exhibits high bias (such as the DM method or the direct estimation of $Q^\pi$) or high variance (such as the DR method).

## 5.3. Choice of Convex Function $f$

We analyze the choice of the convex function $f$. We consider a simple continuous grid task where an agent may move left, right, up, or down and is rewarded for reaching the bottom right corner of a square room. We plot the estimation

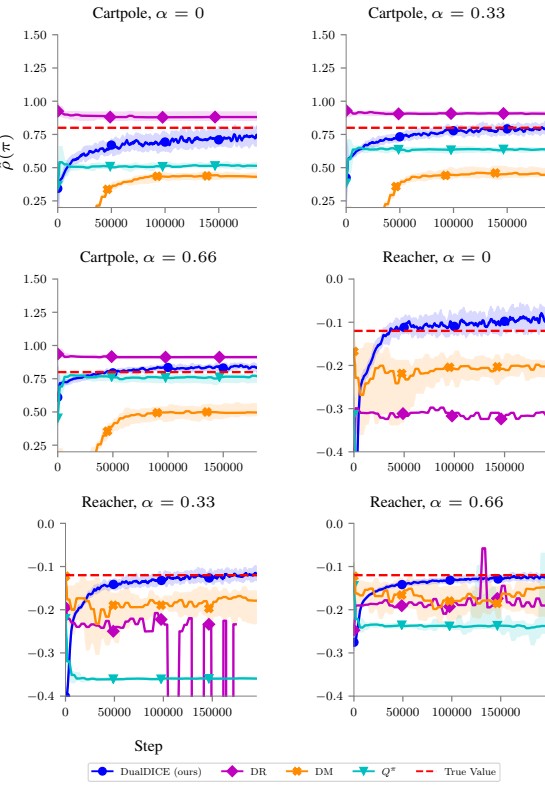

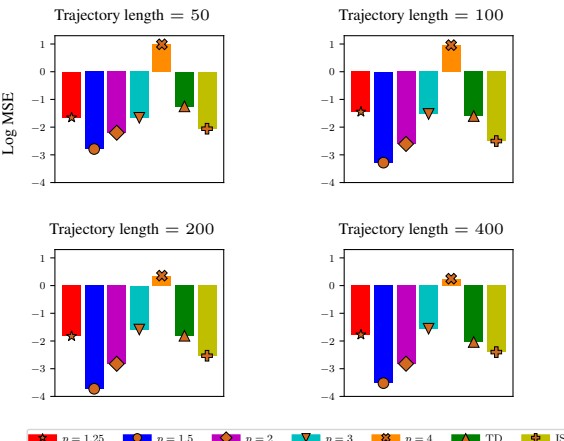

*Figure 4.* We compare the OPE error when using different forms of $f$ to estimate stationary distribution ratios with function approximation, which are then applied to OPE on a simple continuous grid task. In this setting, optimization stability is crucial, and this heavily depends on the form of the convex function $f$. We plot the results of using $f(x) = \frac{1}{p}|x|^p$ for $p \in [1.25, 1.5, 2, 3, 4]$. We also show the results of TD and IS methods on this task for comparison. We find that $p = 1.5$ consistently performs the best, often providing significantly better results.

and stochastic optimization.

Future work includes (1) incorporating the DualDICE algorithm into off-policy training, (2) further understanding the effects of $f$ on the performance of DualDICE (in terms of approximation error of the distribution corrections), and (3) evaluating DualDICE on real-world off-policy evaluation tasks.

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

Boucheron, Stephane, Lugosi, Gabor, and Massart, Pascal.

*Figure 3.* We perform OPE on control tasks using our method compared to a number of additional baselines: doubly-robust (DR), in which one learns a value function in order to reduce the variance of an IS estimate of the evaluation; direct method (DM), in which one learns a model of the dynamics and reward of the environment and performs Monte Carlo rollouts using the model in order to estimate the value of the target policy; and $Q^\pi$, in which one learns $Q^\pi$ values via Bellman error minimization over the off-policy data, and uses the initial values $(1-\gamma) \cdot Q^\pi(s_0, a_0)$ as estimates of the policy value (these estimates are below $-0.4$ for Reacher, $\alpha = 0$).

errors of using DualDICE for off-policy policy evaluation on this task, comparing against different choices of convex functions of the form $f(x) = \frac{1}{p}|x|^p$. Interestingly, although the choice of $f(x) = \frac{1}{2}x^2$ is most natural, we find the empirically best performing choice to be $f(x) = \frac{2}{3}|x|^{3/2}$. Thus, this is the form of $f$ we used in our experiments for Figure 2 and Figure 3.

## 6. Conclusions

We have presented DualDICE, a method for estimating off-policy stationary distribution corrections. Compared to previous work, our method is agnostic to knowledge of the behavior policy used to collect the off-policy data and avoids the use of importance weights in its losses. These advantages have a profound empirical effect: our method provides significantly better estimates compared to TD methods, especially in settings which require function approximation

*Concentration Inequalities: A Nonasymptotic Theory of Independence*. Oxford University Press, 2016.

Brockman, Greg, Cheung, Vicki, Pettersson, Ludwig, Schneider, Jonas, Schulman, John, Tang, Jie, and Zaremba, Wojciech. Openai gym. *arXiv preprint arXiv:1606.01540*, 2016.

Cheng, Kuang Fu, Chu, Chih-Kang, et al. Semiparametric density estimation under a two-sample density ratio model. *Bernoulli*, 10(4):583–604, 2004.

Dai, Bo, He, Niao, Pan, Yunpeng, Boots, Byron, and Song, Le. Learning from conditional distributions via dual embeddings. *arXiv preprint arXiv:1607.04579*, 2016.

Dai, Bo, Shaw, Albert, Li, Lihong, Xiao, Lin, He, Niao, Liu, Zhen, Chen, Jianshu, and Song, Le. Sbeed: Convergent reinforcement learning with nonlinear function approximation. *arXiv preprint arXiv:1712.10285*, 2017.

Dietterich, Thomas G. Hierarchical reinforcement learning with the MAXQ value function decomposition. *Journal of Artificial Intelligence Research*, 13:227–303, 2000.

Du, Simon S, Chen, Jianshu, Li, Lihong, Xiao, Lin, and Zhou, Dengyong. Stochastic variance reduction methods for policy evaluation. In *Proceedings of the 34th International Conference on Machine Learning-Volume 70*, pp. 1049–1058. JMLR. org, 2017.

Dudík, Miroslav, Langford, John, and Li, Lihong. Doubly robust policy evaluation and learning. *arXiv preprint arXiv:1103.4601*, 2011.

Farajtabar, Mehrdad, Chow, Yinlam, and Ghavamzadeh, Mohammad. More robust doubly robust off-policy evaluation. *arXiv preprint arXiv:1802.03493*, 2018.

Fonteneau, Raphael, Murphy, Susan A., Wehenkel, Louis, and Ernst, Damien. Batch mode reinforcement learning based on the synthesis of artificial trajectories. *Annals of Operations Research*, 208(1):383–416, 2013.

Gao, Jianfeng, Galley, Michel, and Li, Lihong. Neural approaches to Conversational AI. *Foundations and Trends in Information Retrieval*, 13(2–3):127–298, 2019.

Gelada, Carles and Bellemare, Marc G. Off-policy deep reinforcement learning by bootstrapping the covariate shift. *AAAI*, 2018.

Gretton, Arthur, Smola, Alex J, Huang, Jiayuan, Schmittfull, Marcel, Borgwardt, Karsten M, and Schöllkopf, Bernhard. Covariate shift by kernel mean matching. In *Dataset shift in machine learning*, pp. 131–160. MIT Press, 2009.

Hallak, Assaf and Mannor, Shie. Consistent on-line off-policy evaluation. In *Proceedings of the 34th International Conference on Machine Learning-Volume 70*, pp. 1372–1383. JMLR. org, 2017.

Hastings, W Keith. Monte carlo sampling methods using markov chains and their applications. 1970.

Haussler, David. Sphere packing numbers for subsets of the boolean n-cube with bounded vapnik-chervonenkis dimension. *Journal of Combinatorial Theory, Series A*, 69(2):217–232, 1995.

Jiang, Nan and Li, Lihong. Doubly robust off-policy value evaluation for reinforcement learning. In *Proceedings of the 33rd International Conference on Machine Learning*, pp. 652–661, 2016.

Kanamori, Takafumi, Hido, Shohei, and Sugiyama, Masashi. A least-squares approach to direct importance estimation. *Journal of Machine Learning Research*, 10(Jul):1391–1445, 2009.

Lazaric, Alessandro, Ghavamzadeh, Mohammad, and Munos, Rémi. Finite-sample analysis of least-squares policy iteration. *Journal of Machine Learning Research*, 13(Oct):3041–3074, 2012.

Li, Jiwei, Monroe, Will, Ritter, Alan, Galley, Michel, Gao, Jianfeng, and Jurafsky, Dan. Deep reinforcement learning for dialogue generation. *arXiv preprint arXiv:1606.01541*, 2016.

Li, Lihong, Chu, Wei, Langford, John, and Wang, Xuanhui. Unbiased offline evaluation of contextual-bandit-based news article recommendation algorithms. In *Proceedings of the fourth ACM international conference on Web search and data mining*, pp. 297–306. ACM, 2011.

Li, Lihong, Munos, Rémi, and Szepesvàri, Csaba. Toward minimax off-policy value estimation. In *Proceedings of the 18th International Conference on Artificial Intelligence and Statistics*, pp. 608–616, 2015.

Liu, Qiang, Li, Lihong, Tang, Ziyang, and Zhou, Dengyong. Breaking the curse of horizon: Infinite-horizon off-policy estimation. In *Advances in Neural Information Processing Systems*, pp. 5356–5366, 2018.

Liu, Yao, Swaminathan, Adith, Agarwal, Alekh, and Brunskill, Emma. Off-policy policy gradient with state distribution correction. In *Proceedings of the Thirty-Fifth Conference on Uncertainty in Artificial Intelligence*, 2019. To appear.

Mahmood, A., van Hasselt, H., and Sutton, R. Weighted importance sampling for off-policy learning with linear

function approximation. In *Proceedings of the 27th International Conference on Neural Information Processing Systems*, 2014.

Mandel, Travis, Liu, Yun-En, Levine, Sergey, Brunskill, Emma, and Popovic, Zoran. Offline policy evaluation across representations with applications to educational games. In *Proceedings of the 2014 international conference on Autonomous agents and multi-agent systems*, pp. 1077–1084. International Foundation for Autonomous Agents and Multiagent Systems, 2014.

Mnih, Volodymyr, Kavukcuoglu, Koray, Silver, David, Graves, Alex, Antonoglou, Ioannis, Wierstra, Daan, and Riedmiller, Martin. Playing atari with deep reinforcement learning. *arXiv preprint arXiv:1312.5602*, 2013.

Murphy, Susan A, van der Laan, Mark J, Robins, James M, and Group, Conduct Problems Prevention Research. Marginal mean models for dynamic regimes. *Journal of the American Statistical Association*, 96(456):1410–1423, 2001.

Nguyen, XuanLong, Wainwright, Martin J, and Jordan, Michael I. Estimating divergence functionals and the likelihood ratio by convex risk minimization. *IEEE Transactions on Information Theory*, 56(11):5847–5861, 2010.

Paduraru, C. *Off-policy Evaluation in Markov Decision Processes*. PhD thesis, McGill University, 2013.

Pollard, D. *Convergence of Stochastic Processes*. David Pollard, 1984.

Precup, D., Sutton, R., and Singh, S. Eligibility traces for off-policy policy evaluation. In *Proceedings of the 17th International Conference on Machine Learning*, pp. 759–766, 2000.

Precup, D., Sutton, R., and Dasgupta, S. Off-policy temporal difference learning with function approximation. In *Proceedings of the 18th International Conference on Machine Learning*, pp. 417–424, 2001.

Precup, Doina. Eligibility traces for off-policy policy evaluation. *Computer Science Department Faculty Publication Series*, pp. 80, 2000.

Puterman, Martin L. Markov decision processes: Discrete stochastic dynamic programming. 1994.

Qin, Jing. Inferences for case-control and semiparametric two-sample density ratio models. *Biometrika*, 85(3):619–630, 1998.

Rockafellar, R Tyrrell and Wets, Roger J-B. *Variational analysis*, volume 317. Springer Science & Business Media, 2009.

Rockafellar, Ralph Tyrell. *Convex analysis*. Princeton university press, 2015.

Shapiro, Alexander, Dentcheva, Darinka, and Ruszczyński, Andrzej. *Lectures on stochastic programming: modeling and theory*. SIAM, 2009.

Sugiyama, Masashi, Suzuki, Taiji, Nakajima, Shinichi, Kashima, Hisashi, von Bünau, Paul, and Kawanabe, Motoaki. Direct importance estimation for covariate shift adaptation. *Annals of the Institute of Statistical Mathematics*, 60(4):699–746, 2008.

Sugiyama, Masashi, Suzuki, Taiji, and Kanamori, Takafumi. *Density ratio estimation in machine learning*. Cambridge University Press, 2012.

Sutton, Richard S and Barto, Andrew G. *Introduction to reinforcement learning*, volume 135.

Sutton, Richard S, Szepesvári, Csaba, Geramifard, Alborz, and Bowling, Michael. Dyna-style planning with linear function approximation and prioritized sweeping. In *Proceedings of the Twenty-Fourth Conference on Uncertainty in Artificial Intelligence*, pp. 528–536. AUAI Press, 2008.

Sutton, Richard S, Mahmood, A Rupam, and White, Martha. An emphatic approach to the problem of off-policy temporal-difference learning. *The Journal of Machine Learning Research*, 17(1):2603–2631, 2016.

Swaminathan, A., Krishnamurthy, A., Agarwal, A., Dudík, M., Langford, J., Jose, D., and Zitouni, I. Off-policy evaluation for slate recommendation. In *Proceedings of the 31st International Conference on Neural Information Processing Systems*, pp. 3635–3645, 2017.

Thomas, P. and Brunskill, E. Data-efficient off-policy policy evaluation for reinforcement learning. In *Proceedings of the 33rd International Conference on Machine Learning*, pp. 2139–2148, 2016.

Thomas, P., Theocharous, G., and Ghavamzadeh, M. High confidence off-policy evaluation. In *Proceedings of the 29th Conference on Artificial Intelligence*, 2015.

Wang, Yu-Xiang, Agarwal, Alekh, and Dudik, Miroslav. Optimal and adaptive off-policy evaluation in contextual bandits. In *Proceedings of the 34th International Conference on Machine Learning-Volume 70*, pp. 3589–3597. JMLR. org, 2017.

Yu, Bin. Rates of convergence for empirical processes of stationary mixing sequences. *The Annals of Probability*, pp. 94–116, 1994.

# A. Pseudocode

---

**Algorithm 1** DualDICE

---

**Inputs**: Convex function $f$ and its Fenchel conjugate $f^*$, off-policy data $\hat{\mathcal{D}} = \{(s^{(i)}, a^{(i)}, r^{(i)}, s'^{(i)})\}_{i=1}^N$, sampled initial states $\hat{\beta} = \{s_0^{(i)}\}_{i=1}^M$, target policy $\pi$, networks $\nu_{\theta_1}(\cdot, \cdot), \zeta_{\theta_2}(\cdot, \cdot)$, learning rates $\eta_\nu, \eta_\zeta$, number of iterations $T$, batch size $B$.

**for** $t = 1, \ldots, T$ **do**

    Sample batch $\{(s^{(i)}, a^{(i)}, r^{(i)}, s'^{(i)})\}_{i=1}^B$ from $\hat{\mathcal{D}}$.

    Sample batch $\{s_0^{(i)}\}_{i=1}^B$ from $\hat{\beta}$.

    Sample actions $a'^{(i)} \sim \pi(s'^{(i)})$, for $i = 1, \ldots, B$.

    Sample actions $a_0^{(i)} \sim \pi(s_0^{(i)})$, for $i = 1, \ldots, B$.

    Compute empirical loss $\hat{J} = \frac{1}{B} \sum_{i=1}^B (\nu_{\theta_1}(s^{(i)}, a^{(i)}) - \nu_{\theta_1}(s'^{(i)}, a'^{(i)})) \zeta_{\theta_2}(s^{(i)}, a^{(i)}) - f^*(\zeta_{\theta_2}(s^{(i)}, a^{(i)})) - (1 - \gamma)\nu_{\theta_1}(s_0^{(i)}, a_0^{(i)})$.

    Update $\theta_1 \leftarrow \theta_1 - \eta_\nu \nabla_{\theta_1} \hat{J}$.

    Update $\theta_2 \leftarrow \theta_2 + \eta_\zeta \nabla_{\theta_2} \hat{J}$.

**end for**

**Return** $\zeta_{\theta_2}(\cdot, \cdot)$.

---

# B. Experimental Details

### B.1. Taxi

For the Taxi domain, we follow the same protocol as used in (Liu et al., 2018). In this tabular, exact solve setting, the TD methods (Gelada & Bellemare, 2018) are equivalent to their kernel-based TD method. We fix $\gamma$ to $0.995$. The behavior and target policies are also taken from (Liu et al., 2018) (referred in their work as the behavior policy for $\alpha = 0$).

In this setting, we solve for the optimal empirical $\nu$ exactly using matrix operations. Since (Liu et al., 2018) perform a similar exact solve for $|S|$ variables $w_{\pi/\mu}(s)$, for better comparison we also perform our exact solve with respect to $|S|$ variables $\nu(s)$. Specifically, one may follow the same derivations for DualDICE with respect to learning $w_{\pi/\mu}$. The final objective will require knowledge of the importance weights $\pi(a|s)/\mu(a|s)$.

### B.2. Control Tasks

We use the Cartpole and Reacher tasks as given by OpenAI Gym (Brockman et al., 2016). In these tasks we use COP-TD (Gelada & Bellemare, 2018) for the TD method ((Liu et al., 2018) requires a proper kernel, which is not readily available for these tasks). When assuming an unknown $\mu$, we learn a neural network policy $\hat{\mu}$ using behavior cloning, and use its probabilities for computing importance weights $\pi(a|s)/\mu(a|s)$. All neural networks are feed-forward with two hidden layers of dimension $64$ and `tanh` activations.

We modify the Cartpole task to be infinite horizon: We use the same dynamics as in the original task but change the reward to be $-1$ if the original task returns a termination (when the pole falls below some threshold) and $1$ otherwise. We train a policy on this task until convergence. We then define the target policy $\pi$ as a weighted combination of this pre-trained policy (weight $0.7$) and a uniformly random policy (weight $0.3$). The behavior policy $\mu$ for a specific $0 \le \alpha \le 1$ is taken to be a weighted combination of the pre-trained policy (weight $0.55 + 0.15\alpha$) and a uniformly random policy (weight $0.45 - 0.15\alpha$). We use $\gamma = 0.99$, which yields an average step reward of $\approx 0.8$ for $\pi$ and $\approx 0.1$ for $\mu$ with $\alpha = 0$. We generate an off-policy dataset by running the behavior policy for $200$ epsiodes, each of length $250$ steps. We train each stationary distribution correction estimation method using the Adam optimizer with batches of size $2048$ and learning rates chosen using a hyperparameter search (the optimal learning rate found for either method was $\approx 0.003$).

For the Reacher task, we train a deterministic policy until convergence. We define the target policy $\pi$ as a Gaussian with mean given by the pre-trained policy and standard deviation given by $0.1$. The behavior policy $\mu$ for a specific $0 \le \alpha \le 1$ is taken to be a Gaussian with mean given by the pre-trained policy and standard deviation given by $0.4 - 0.3\alpha$. We use $\gamma = 0.99$, which yields an average step reward of $\approx -0.12$ for $\pi$ and $\approx -0.50$ for $\mu$ with $\alpha = 0$. We generate an off-policy dataset by running the behavior policy for $1000$ epsiodes, each of length $40$ steps. We train each stationary

distribution correction estimation method using the Adam optimizer with batches of size 2048 and learning rates chosen using a hyperparameter search (the optimal learning rate found for either method was $\approx 0.0001$).

### B.3. Continuous Grid

For this task, we create a $10 \times 10$ grid which the agent can traverse by moving left/right/up/down. The observations are the $x, y$ coordinates of the square the agent is on. The reward at each step is given by $\exp\{-0.2|x-9| - 0.2|y-9|\}$. We use $\gamma = 0.995$. The target policy $\pi$ is taken to be the optimal policy for this task plus 0.1 weight on uniform exploration. The behavior policy $\mu$ is taken to be the optimal policy plus 0.7 weight on uniform exploration. We train using batches of size the Adam optimizer with batches of size 512 and learning rates 0.001 for $\nu$ and 0.0001 for $\zeta$.

## C. Proofs

We provide the proof for Theorem 2. We first decompose the error in Section C.1. Then, we analyze the statistical error and optimization error in Section C.2 and Section C.4, respectively. The total error will be discussed in C.3.

Although the proposed estimator can use any general convex function $f$, as a first step towards a more complete theoretical understanding, we consider the special case of $f(x) = \frac{1}{2}x^2$. Clearly, $f(\cdot)$ now is $\eta$-strongly convex with $\eta = 1$. Under Assumption 1, we need only consider $\|\nu\|_\infty \leq C$, which implies that $\|\nu - \mathcal{B}^\pi \nu\|_\infty \leq \frac{1+\gamma}{1-\gamma}C$, and that $f(x)$ is $\kappa$-Lipschitz continuous with $\kappa = \frac{1+\gamma}{1-\gamma}C$. Similarly, $f^*(y) = \frac{1}{2}y^2$ is $L$-Lipschitz continuous with $L = C$ on $\|w\|_\infty \leq C$. The following assumption will be needed.

**Assumption 3** (MDP regularity). *We assume the observed reward is uniformly bounded*, i.e., $\|\hat{r}(s,a)\|_\infty \leq C_r$ *for some constant $C_r > 0$. It follows that the reward's mean and variance are both bounded in $[-C_r, C_r]$.*

For convenience, the objective function of DualDICE is repeated here:

$$J(\nu, \zeta) = \mathbb{E}_{(s,a,s') \sim d^{\mathcal{D}}, a' \sim \pi(s')} \left[ (\nu(s,a) - \gamma\nu(s',a'))\zeta(s,a) - \zeta(s,a)^2/2 \right] \\ - (1-\gamma) \mathbb{E}_{s_0 \sim \beta, a_0 \sim \pi(s_0)} \left[ \nu(s_0, a_0) \right].$$

We will also make use of the objective in the form prior to introduction of $\zeta$, which we denote as $J(\nu)$:

$$J(\nu) = \frac{1}{2} \mathbb{E}_{(s,a) \sim d^{\mathcal{D}}} \left[ (\nu - \mathcal{B}^\pi \nu)(s,a)^2 \right] - (1-\gamma) \mathbb{E}_{s_0 \sim \beta, a_0 \sim \pi(s_0)} \left[ \nu(s_0, a_0) \right].$$

Let $\hat{J}(\nu, \zeta)$ denotes the empirical surrogate of $J(\nu, \zeta)$ with optimal solution as $(\hat{\nu}^*, \hat{\zeta}^*)$. We denote $\nu_{\mathcal{F}}^* = \arg\min_{\nu \in \mathcal{F}} J(\nu)$ and $\nu^* = \arg\min_{\nu \in S \times A \to \mathbb{R}} J(\nu)$. We denote $L(\nu) = \max_{\zeta \in \mathcal{H}} J(\nu, \zeta)$ and $\hat{L}(\nu) = \max_{\zeta \in \mathcal{H}} \hat{J}(\nu, \zeta)$ as the primal objectives, and $\ell(\zeta) = \min_{\nu \in \mathcal{F}} J(\nu, \zeta)$, $\hat{\ell}(\zeta) = \min_{\nu \in \mathcal{F}} \hat{J}(\nu, \zeta)$ as the dual objectives. We apply some optimization algorithm $OPT$ for optimizing $\hat{J}(\nu, \zeta)$ with samples $\{s_i, a_i, r_i, s_i'\}_{i=1}^N \sim d^{\mathcal{D}}$, $\{s_0^i\}_{i=1}^N \sim \beta$, and target actions $a_i' \sim \pi(s_i'), a_0^i \sim \pi(s_0^i)$ for $i = 1, \ldots, N$. We denote the outputs of $OPT$ by $(\hat{\nu}, \hat{\zeta})$.

### C.1. Error Decomposition

Let

$$\overline{R}(s,a) = \mathbb{E}_{\cdot|s,a}[r].$$

Observe that

$$\rho(\pi) = \mathbb{E}_{d^{\mathcal{D}}} \left[ w_{\pi/\mathcal{D}}(s,a) \cdot \overline{R}(s,a) \right].$$

We begin by considering the estimation error induced by using $(\hat{\nu} - \hat{\mathcal{B}}^\pi \hat{\nu})(s,a)$ as estimates of $w_{\pi/\mathcal{D}}(s,a)$, where $\hat{\mathcal{B}}^\pi$ denotes the empirical Bellman backup with respect to samples from $d^{\mathcal{D}}, \pi$. We will subsequently reconcile this with the true implementation of DualDICE, which uses $\hat{\zeta}(s,a)$ as estimates of $w_{\pi/\mathcal{D}}(s,a)$.

The mean squared error of the policy value estimate when using $(\hat{\nu} - \hat{\mathcal{B}}^\pi \hat{\nu})(s,a)$ in place of $w_{\pi/\mathcal{D}}(s,a)$ can be decomposed

as

$$\left(\hat{\mathbb{E}}_{d^{\mathcal{D}}}\left[\left(\hat{\nu}-\hat{\mathcal{B}}^{\pi}\hat{\nu}\right)(s,a)\cdot r\right]-\mathbb{E}_{d^{\mathcal{D}}}\left[w_{\pi/\mathcal{D}}(s,a)\cdot\overline{R}(s,a)\right]\right)^2 \tag{19}$$

$$= \left(\hat{\mathbb{E}}_{d^{\mathcal{D}}}\left[\left(\hat{\nu}-\hat{\mathcal{B}}^{\pi}\hat{\nu}\right)(s,a)\cdot r\right]-\hat{\mathbb{E}}_{d^{\mathcal{D}}}\left[\left(\hat{\nu}-\hat{\mathcal{B}}^{\pi}\hat{\nu}\right)(s,a)\cdot\overline{R}(s,a)\right]\right. \tag{20}$$

$$+\hat{\mathbb{E}}_{d^{\mathcal{D}}}\left[\left(\hat{\nu}-\hat{\mathcal{B}}^{\pi}\hat{\nu}\right)(s,a)\cdot\overline{R}(s,a)\right]-\hat{\mathbb{E}}_{d^{\mathcal{D}}}\left[\left(\hat{\nu}^*-\hat{\mathcal{B}}^{\pi}\hat{\nu}^*\right)(s,a)\cdot\overline{R}(s,a)\right]$$

$$+\hat{\mathbb{E}}_{d^{\mathcal{D}}}\left[\left(\hat{\nu}^*-\hat{\mathcal{B}}^{\pi}\hat{\nu}^*\right)(s,a)\cdot\overline{R}(s,a)\right]-\mathbb{E}_{d^{\mathcal{D}}}\left[w_{\pi/\mathcal{D}}(s,a)\cdot\overline{R}(s,a)\right]\Bigg)^2$$

$$\leq 4\underbrace{\left(\hat{\mathbb{E}}_{d^{\mathcal{D}}}\left[\left(\hat{\nu}-\hat{\mathcal{B}}^{\pi}\hat{\nu}\right)(s,a)\cdot r\right]-\hat{\mathbb{E}}_{d^{\mathcal{D}}}\left[\left(\hat{\nu}-\hat{\mathcal{B}}^{\pi}\hat{\nu}\right)(s,a)\cdot\overline{R}(s,a)\right]\right)^2}_{\epsilon_r} \tag{21}$$

$$+4\underbrace{\left(\hat{\mathbb{E}}_{d^{\mathcal{D}}}\left[\left(\hat{\nu}-\hat{\mathcal{B}}^{\pi}\hat{\nu}\right)(s,a)\cdot\overline{R}(s,a)\right]-\hat{\mathbb{E}}_{d^{\mathcal{D}}}\left[\left(\hat{\nu}^*-\hat{\mathcal{B}}^{\pi}\hat{\nu}^*\right)(s,a)\cdot\overline{R}(s,a)\right]\right)^2}_{\epsilon_1} \tag{22}$$

$$+4\underbrace{\left(\hat{\mathbb{E}}_{d^{\mathcal{D}}}\left[\left(\hat{\nu}^*-\hat{\mathcal{B}}^{\pi}\hat{\nu}^*\right)(s,a)\cdot\overline{R}(s,a)\right]-\mathbb{E}_{d^{\mathcal{D}}}\left[w_{\pi/\mathcal{D}}(s,a)\cdot\overline{R}(s,a)\right]\right)^2}_{\epsilon_2}. \tag{23}$$

The first term, $\epsilon_r$, is induced by the randomness in observed reward, and we have

$$\epsilon_r \leq \left(\hat{\mathbb{E}}_{d^{\mathcal{D}}}\left[\left(\hat{\nu}-\hat{\mathcal{B}}^{\pi}\hat{\nu}\right)(s,a)\cdot(\hat{r}(s,a)-r(s,a))\right]\right)^2 \leq \left(\frac{1+\gamma}{1-\gamma}\right)^2 C^2 \left(\hat{\mathbb{E}}_{d^{\mathcal{D}}}[\hat{r}(s,a)]-\hat{\mathbb{E}}_{d^{\mathcal{D}}}[r(s,a)]\right)^2,$$

which will be discussed in section C.2.

We consider the $\epsilon_1$ as

$$\epsilon_1 \leq C_r^2 \left\|\left(\hat{\nu}-\hat{\mathcal{B}}^{\pi}\hat{\nu}\right)-\left(\hat{\nu}^*-\hat{\mathcal{B}}^{\pi}\hat{\nu}^*\right)\right\|_{\hat{\mathcal{D}}}^2 \leq C_r^2 \left(\underbrace{\left\|\hat{\zeta}-\hat{\zeta}^*\right\|_{\hat{\mathcal{D}}}^2+\left\|\left(\hat{\nu}^*-\hat{\mathcal{B}}^{\pi}\hat{\nu}^*\right)-\left(\hat{\nu}-\hat{\mathcal{B}}^{\pi}\hat{\nu}\right)\right\|_{\hat{\mathcal{D}}}^2}_{\hat{\epsilon}_{opt}}\right)$$

which is the error induced by optimization $OPT$.

For the last term $\epsilon_2$, we have

$$\epsilon_2 \leq 2\underbrace{\left(\hat{\mathbb{E}}_{d^{\mathcal{D}}}\left[\left(\hat{\nu}^*-\hat{\mathcal{B}}^{\pi}\hat{\nu}^*\right)(s,a)\cdot r(s,a)\right]-\mathbb{E}_{d^{\mathcal{D}}}\left[(\hat{\nu}^*-\mathcal{B}^{\pi}\hat{\nu}^*)(s,a)\cdot r(s,a)\right]\right)^2}_{\epsilon_{stat}}$$

$$+2\left(\mathbb{E}_{d^{\mathcal{D}}}\left[(\hat{\nu}^*-\mathcal{B}^{\pi}\hat{\nu}^*)(s,a)\cdot r(s,a)\right]-\mathbb{E}_{d^{\mathcal{D}}}\left[w_{\pi/\mathcal{D}}(s,a)\cdot r(s,a)\right]\right)^2$$

$$\leq 2\epsilon_{stat}+2\left(\mathbb{E}_{d^{\mathcal{D}}}\left[(\hat{\nu}^*-\mathcal{B}^{\pi}\hat{\nu}^*)(s,a)\cdot r(s,a)\right]-\mathbb{E}_{d^{\mathcal{D}}}\left[(\nu^*-\mathcal{B}^{\pi}\nu^*)(s,a)\cdot r(s,a)\right]\right)^2. \quad \text{(due to equation 17)}$$

For the first term $\epsilon_{stat}$, which is due to finite samples, we will bound in section C.2.

For the second term, we have

$$\left(\mathbb{E}_{d^{\mathcal{D}}}\left[(\hat{\nu}^*-\mathcal{B}^{\pi}\hat{\nu}^*)(s,a)\cdot r(s,a)\right]-\mathbb{E}_{d^{\mathcal{D}}}\left[(\nu^*-\mathcal{B}^{\pi}\nu^*)(s,a)\cdot r(s,a)\right]\right)^2$$

$$\leq \mathbb{E}_{d^{\mathcal{D}}}\left[r(s,a)^2\cdot((\hat{\nu}^*-\mathcal{B}^{\pi}\hat{\nu}^*)(s,a)-(\nu^*-\mathcal{B}^{\pi}\nu^*)(s,a))^2\right]$$

$$\leq C_r^2 \left\|(\hat{\nu}^*-\mathcal{B}^{\pi}\hat{\nu}^*)-(\nu^*-\mathcal{B}^{\pi}\nu^*)\right\|_{\mathcal{D}}^2$$

$$\leq \frac{2C_r^2}{\eta}\left(J(\hat{\nu}^*)-J(\nu^*)\right),$$

where the last inequality comes from the $\eta$-strongly convexity of $f$ and the optimality of $\nu^*$.

We then consider the error between $J(\hat{\nu}^*)$ and $J(\nu^*)$, which can be decomposed as

$$
\begin{aligned}
J(\hat{\nu}^*) - J(\nu^*) &= J(\hat{\nu}^*) - J(\nu_{\mathcal{F}}^*) + J(\nu_{\mathcal{F}}^*) - J(\nu^*) \\
&= J(\hat{\nu}^*) - L(\hat{\nu}^*) + L(\hat{\nu}^*) - L(\nu_{\mathcal{F}}^*) + L(\nu_{\mathcal{F}}^*) - J(\nu_{\mathcal{F}}^*) + J(\nu_{\mathcal{F}}^*) - J(\nu^*).
\end{aligned}
$$

We bound this expression term-by-term from the right. For the term $J(\nu_{\mathcal{F}}^*) - J(\nu^*)$, we have

$$
\begin{aligned}
J(\nu_{\mathcal{F}}^*) - J(\nu^*) &= \mathbb{E}_{\mathcal{D}}\left[f(\nu_{\mathcal{F}}^* - \mathcal{B}^\pi \nu_{\mathcal{F}}^*) - f(\nu^* - \mathcal{B}^\pi \nu^*)\right] - \mathbb{E}_{\beta\pi}\left[\nu_{\mathcal{F}}^* - \nu^*\right] \\
&\leq \kappa\|\nu_{\mathcal{F}}^* - \nu^*\|_{\mathcal{D},1} + \kappa\|\mathcal{B}^\pi(\nu_{\mathcal{F}}^* - \nu^*)\|_{\mathcal{D},1} + \|\nu_{\mathcal{F}}^* - \nu^*\|_{\beta\pi,1} \\
&\leq \max\left(\kappa + \kappa\|\mathcal{B}^\pi\|_{\mathcal{D},1}, 1\right)\left(\|\nu_{\mathcal{F}}^* - \nu^*\|_{\mathcal{D},1} + \|\nu_{\mathcal{F}}^* - \nu^*\|_{\beta\pi,1}\right) \\
&\leq \max\left(\kappa + \kappa\|\mathcal{B}^\pi\|_{\mathcal{D},1}, 1\right) \cdot \epsilon_{approx}(\mathcal{F}),
\end{aligned}
$$

where $\epsilon_{approx}(\mathcal{F}) := \sup_{\nu \in S \times A \to \mathbb{R}} \inf_{\nu \in \mathcal{F}}\left(\|\nu_{\mathcal{F}} - \nu\|_{\mathcal{D},1} + \|\nu_{\mathcal{F}} - \nu\|_{\beta\pi,1}\right)$, due to the approximation with $\mathcal{F}$ for $\nu$.

For the term $L(\nu_{\mathcal{F}}^*) - J(\nu_{\mathcal{F}}^*)$, we have by definition that

$$
L(\nu_{\mathcal{F}}^*) - J(\nu_{\mathcal{F}}^*) = \max_{\zeta \in \mathcal{H}} J(\nu_{\mathcal{F}}^*, \zeta) - \max_{\zeta \in S \times A \to \mathbb{R}} J(\nu_{\mathcal{F}}^*, \zeta) \leq 0
$$

For the term $L(\hat{\nu}^*) - L(\nu_{\mathcal{F}}^*)$,

$$
\begin{aligned}
L(\hat{\nu}^*) - L(\nu_{\mathcal{F}}^*) &= L(\hat{\nu}^*) - \hat{L}(\hat{\nu}^*) + \hat{L}(\hat{\nu}^*) - \hat{L}(\nu_{\mathcal{F}}^*) + \hat{L}(\nu_{\mathcal{F}}^*) - L(\nu_{\mathcal{F}}^*) \\
&\leq L(\hat{\nu}^*) - \hat{L}(\hat{\nu}^*) + \hat{L}(\nu_{\mathcal{F}}^*) - L(\nu_{\mathcal{F}}^*) \\
&\leq 2\sup_{\nu \in \mathcal{F}}\left|L(\nu) - \hat{L}(\nu)\right| \\
&= 2\sup_{\nu \in \mathcal{F}}\left|\max_{\zeta \in \mathcal{H}} J(\nu, \zeta) - \max_{\zeta \in \mathcal{H}} \hat{J}(\nu, \zeta)\right| \\
&\leq 2\sup_{\nu \in \mathcal{F}, \zeta \in \mathcal{H}}\left|\hat{J}(\nu, \zeta) - J(\nu, \zeta)\right| \\
&= 2 \cdot \epsilon_{est}(\mathcal{F}),
\end{aligned}
$$

where in the first inequality we have used the fact that $\hat{L}(\hat{\nu}^*) - \hat{L}(\nu_{\mathcal{F}}^*) \leq 0$ due to the optimality of $\hat{\nu}^*$, and in the last step $\epsilon_{est}(\mathcal{F}) := \sup_{\nu \in \mathcal{F}, \zeta \in \mathcal{H}}\left|\hat{J}(\nu, \zeta) - J(\nu, \zeta)\right|$.

For the term $J(\hat{\nu}^*) - L(\hat{\nu}^*)$, we have

$$
\begin{aligned}
J(\hat{\nu}^*) - L(\hat{\nu}^*) &= \max_{\zeta \in S \times A \to \mathbb{R}} J(\hat{\nu}^*, \zeta) - \max_{\zeta \in \mathcal{H}} J(\hat{\nu}^*, \zeta) \\
&\leq \left(L + \frac{1+\gamma}{1-\gamma}C\right)\underbrace{\|\zeta_{\mathcal{H}}^* - \zeta^*\|_{\mathcal{D},1}}_{\leq \epsilon_{approx}(\mathcal{H})},
\end{aligned}
$$

where $\epsilon_{approx}(\mathcal{H}) := \sup_{\zeta \in S \times A \to \mathbb{R}} \inf_{\zeta \in \mathcal{H}}\left(\|\zeta_{\mathcal{H}} - \zeta\|_{\mathcal{D},1} + \|\zeta_{\mathcal{H}} - \zeta\|_{\beta\pi,1}\right)$, due to the approximation with $\mathcal{H}$ for $\zeta$.

Finally, we can decompose the squared error as

$$
\begin{aligned}
\left(\hat{\mathbb{E}}_{d^{\mathcal{D}}}\left[\left(\hat{\nu} - \hat{\mathcal{B}}^\pi \hat{\nu}\right)(s,a) \cdot \hat{r}(s,a)\right] - \rho(\pi)\right)^2 \\
\leq \frac{16C_r^2}{\eta}\left(\max\left(\kappa + \kappa\|\mathcal{B}^\pi\|_{\mathcal{D},1}, 1\right)\epsilon_{approx}(\mathcal{F}) + \left(L + \frac{1+\gamma}{1-\gamma}C\right)\epsilon_{approx}(\mathcal{H})\right) \\
+ 4\epsilon_r + 8\epsilon_{stat} + \frac{32C_r^2}{\eta}\epsilon_{est}(\mathcal{F}) + 4\hat{\epsilon}_{opt}. \quad (24)
\end{aligned}
$$

**Remark (Dual OPE estimator):** We now reconcile the above derivations with the use of $\hat{\zeta}(s,a)$ as estimates of $w_{\pi/\mathcal{D}}(s,a)$. Note that in the implementation of DualDICE we use the estimator,

$$\hat{\mathbb{E}}_{d^{\mathcal{D}}}\left[\hat{\zeta}(s,a)\cdot r\right]$$

for off-policy policy evaluation. In this case, the error can be decomposed as

$$\left(\hat{\mathbb{E}}_{d^{\mathcal{D}}}\left[\hat{\zeta}(s,a)\cdot r\right] - \mathbb{E}_{d^{\mathcal{D}}}\left[w_{\pi/\mathcal{D}}(s,a)\cdot\overline{R}(s,a)\right]\right)^2 \tag{25}$$

$$\leq 2\left(\hat{\mathbb{E}}_{d^{\mathcal{D}}}\left[\hat{\zeta}(s,a)\cdot r\right] - \hat{\mathbb{E}}_{d^{\mathcal{D}}}\left[\left(\hat{\nu}-\hat{\mathcal{B}}^{\pi}\hat{\nu}\right)(s,a)\cdot r\right]\right)^2 \tag{26}$$

$$+2\left(\hat{\mathbb{E}}_{d^{\mathcal{D}}}\left[\left(\hat{\nu}-\hat{\mathcal{B}}^{\pi}\hat{\nu}\right)(s,a)\cdot r\right] - \mathbb{E}_{d^{\mathcal{D}}}\left[w_{\pi/\mathcal{D}}(s,a)\cdot\overline{R}(s,a)\right]\right)^2. \tag{27}$$

The second term above is the same as given in equation 19. The first term can be rewritten as,

$$\left(\hat{\mathbb{E}}_{d^{\mathcal{D}}}\left[\hat{\zeta}(s,a)\cdot r\right] - \hat{\mathbb{E}}_{d^{\mathcal{D}}}\left[\left(\hat{\nu}-\hat{\mathcal{B}}^{\pi}\hat{\nu}\right)(s,a)\cdot r\right]\right)^2 \leq C_r^2\left\|\hat{\zeta}-\left(\hat{\nu}-\hat{\mathcal{B}}^{\pi}\hat{\nu}\right)\right\|_{\hat{\mathcal{D}}}^2,$$

which can be bounded as follows:

$$\left\|\hat{\zeta}-\left(\hat{\nu}-\hat{\mathcal{B}}^{\pi}\hat{\nu}\right)\right\|_{\hat{\mathcal{D}}}^2$$

$$= \left\|\hat{\zeta}-\hat{\zeta}^*+\hat{\zeta}^*-\left(\hat{\nu}^*-\hat{\mathcal{B}}^{\pi}\hat{\nu}^*\right)+\left(\hat{\nu}^*-\hat{\mathcal{B}}^{\pi}\hat{\nu}^*\right)-\left(\hat{\nu}-\hat{\mathcal{B}}^{\pi}\hat{\nu}\right)\right\|_{\hat{\mathcal{D}}}^2 \tag{28}$$

$$\leq 4\left\|\hat{\zeta}-\hat{\zeta}^*\right\|_{\hat{\mathcal{D}}}^2+4\left\|\left(\hat{\nu}^*-\hat{\mathcal{B}}^{\pi}\hat{\nu}^*\right)-\left(\hat{\nu}-\hat{\mathcal{B}}^{\pi}\hat{\nu}\right)\right\|_{\hat{\mathcal{D}}}^2+4\left\|\hat{\zeta}^*-\left(\hat{\nu}^*-\hat{\mathcal{B}}^{\pi}\hat{\nu}^*\right)\right\|_{\hat{\mathcal{D}}}^2$$

where the first two terms correspond to optimization error $\hat{\epsilon}_{opt}$, and the last to approximation error due to parametrization.

Specifically, when the output of our algorithm $\hat{\zeta}(s,a) = \left(\hat{\nu}-\hat{\mathcal{B}}^{\pi}\hat{\nu}\right)(s,a)$ for $\forall (s,a)\in\hat{\mathcal{D}}$, the extra term vanishes, and the error is the same as in equation 19.

## C.2. Statistical Error

We analyze the statistical error $\epsilon_r$, $\epsilon_{stat}$ and $\epsilon_{est}(\mathcal{F})$ in this section. We discussed in batch learning setting with *i.i.d.* samples (Sutton et al., 2008). However, by exploiting blocking technique in Proposition 15 of (Yu, 1994), following (Antos et al., 2008; Lazaric et al., 2012; Dai et al., 2017), all the sample complexity we provided can be easily generalized for single $\beta$-mixing sample path, *i.e.*, $\{s_i, a_i, r_i, s_i'\}_{i=1}^N$ is strictly stationary and mixing in an exponential rate with parameter $b, \chi > 0$ if $\beta_m = \mathcal{O}(\exp(-bm^{-\chi}))$, which we omit for the sake of exposition simplicity.

**Bounding $\epsilon_r$.** Recall that $\overline{R}(s,a) = \mathbb{E}_{\cdot|s,a}[r]$, so

$$\mathbb{E}[\epsilon_r] \leq \left(\frac{1+\gamma}{1-\gamma}\right)^2 C^2\mathbb{E}\left[\left(\frac{1}{N}\sum_{i=1}^N r_i - \mathbb{E}\left[\frac{1}{N}\sum_{i=1}^N r_i\right]\right)^2\right]$$

$$= \left(\frac{1+\gamma}{1-\gamma}\right)^2 C^2\mathbb{V}\left(\frac{1}{N}\sum_{i=1}^N r_i\right)$$

$$\leq \frac{1}{N}\left(\frac{1+\gamma}{1-\gamma}\right)^2 C^2\sup_{s,a}\mathbb{V}(r|s,a) = \mathcal{O}\left(\frac{1}{N}\right). \tag{29}$$

Since $r(s,a)$ and $\hat{r}(s,a)$ is bounded, we can also obtain high-probability deviation bounds using standard concentration inequalities (Boucheron et al., 2016).

**Bounding $\epsilon_{est}(\mathcal{F})$.** By definition, we have

$$\epsilon_{est}(\mathcal{F}) = \sup_{\nu \in \mathcal{F}, \zeta \in \mathcal{H}} \left| \hat{J}(\nu, \zeta) - J(\nu, \zeta) \right|,$$

which can be bounded using a covering-number argument outlined below.

We will need Pollard's tail inequality that relates maximum deviation to the covering number of a function class:

**Lemma 4.** *(Pollard, 1984) Let $\mathcal{G}$ be a permissible class of $\mathcal{Z} \to [-M, M]$ functions and $\{Z_i\}_{i=1}^N$ are i.i.d. samples from some distribution. Then, for any given $\epsilon > 0$,*

$$\mathbb{P}\left( \sup_{g \in \mathcal{G}} \left| \frac{1}{N} \sum_{i=1}^N g(Z_i) - \mathbb{E}\left[g(Z)\right] \right| > \epsilon \right) \leq 8\mathbb{E}\left[ \mathcal{N}_1\left( \frac{\epsilon}{8}, \mathcal{G}, \{Z_i\}_{i=1}^N \right) \right] \exp\left( \frac{-N\epsilon^2}{512M^2} \right).$$

The covering number can then be bounded in terms of the function class's pseudo-dimension:

**Lemma 5.** *[Corollary 3, (Haussler, 1995)] For any set $\mathcal{X}$, any points $x^{1:N} \in \mathcal{X}^N$, any class $\mathcal{F}$ of functions on $\mathcal{X}$ taking values in $[0, M]$ with pseudo-dimension $D_{\mathcal{F}} < \infty$, and any $\epsilon > 0$,*

$$\mathcal{N}_1\left( \epsilon, \mathcal{F}, x^{1:N} \right) \leq e\left(D_{\mathcal{F}} + 1\right) \left( \frac{2eM}{\epsilon} \right)^{D_{\mathcal{F}}}.$$

With the above technical lemmas, we are ready to bound $\epsilon_{est}(\mathcal{F})$.

**Lemma 6** (Statistical error $\epsilon_{est}(\mathcal{F})$). *Under Assumption 1, if $f^*$ is L-Lipschitz continuous, with at least probability $1 - \delta$,*

$$\epsilon_{est}(\mathcal{F}) = \mathcal{O}\left( \sqrt{\frac{\log N + \log \frac{1}{\delta}}{N}} \right).$$

*Proof.* Denote $h_{\nu,\zeta}(s, a, s', a', s_0, a_0) = (\nu(s, a) - \gamma\nu(s', a'))\zeta(s, a) - f^*(\zeta(s, a)) - (1 - \gamma)\nu(s_0, a_0)$, we use lemma 4 with $\mathcal{Z} = \underbrace{S \times A \times S \times A}_{d^D\pi} \times \underbrace{S \times A}_{\beta\pi}$, $Z_i = \left(s_i, a_i, s_i', a_i', s_0^i, a_0^i\right)$ and $\mathcal{G} = h_{\mathcal{F} \times \mathcal{H}}$.

We first show that $\forall h_{\nu,\zeta} \in \mathcal{G}$ is bounded. Recall $\nu \in \mathcal{F}$ and $\zeta \in \mathcal{H}$ are bounded by $\frac{1}{1-\gamma}C$ and $C$, then, $h_{\nu,\zeta}$ will be bounded by $M_1 = \frac{1+\gamma}{1-\gamma}C^2 + (1 + L)C + |f^*(0)|$. Specifically,

$$
\begin{aligned}
\|h_{\nu,\zeta}\|_\infty &\leq (1 + \gamma)\|\nu\|_\infty \|\zeta\|_\infty + (1 - \gamma)\|\nu\|_\infty + \|f^*(\zeta)\|_\infty \\
&\leq \frac{1+\gamma}{1-\gamma}C^2 + C + \|f^*(\zeta) - f^*(0)\|_\infty + |f^*(0)| \\
&\leq \frac{1+\gamma}{1-\gamma}C^2 + C + L\|\zeta\|_\infty + |f^*(0)| \\
&\leq \frac{1+\gamma}{1-\gamma}C^2 + C + LC + |f^*(0)|.
\end{aligned}
$$

Thus,

$$\mathbb{P}\left( \sup_{\nu \in \mathcal{F}, \zeta \in \mathcal{H}} \left| \hat{J}(\nu, \zeta) - J(\nu, \zeta) \right| \geq \epsilon \right) = \mathbb{P}\left( \sup_{\nu \in \mathcal{F}, \zeta \in \mathcal{H}} \left| \frac{1}{N} \sum_{i=1}^N h_{\nu,\zeta}(Z_i) - \mathbb{E}\left[h_{\nu,\zeta}\right] \right| \geq \epsilon \right)$$

$$\leq 8\mathbb{E}\left[ \mathcal{N}_1\left( \frac{\epsilon}{8}, \mathcal{G}, \{Z_i\}_{i=1}^N \right) \right] \exp\left( \frac{-N\epsilon^2}{512M_1^2} \right). \quad (30)$$

We bound the distance in $\mathcal{G}$,

$$\frac{1}{N}\sum_{i=1}^{N}|h_{\nu_1,\zeta_1}(Z_i) - h_{\nu_2,\zeta_2}(Z_i)|$$

$$\leq \frac{\left(L + \frac{1+\gamma}{1-\gamma}C\right)}{N}\sum_{i=1}^{N}|\zeta_1(s_i,a_i) - \zeta_2(s_i,a_i)| + \frac{C}{N}\sum_{i=1}^{N}|\nu_1(s_i,a_i) - \nu_2(s_i,a_i)|$$

$$+ \frac{\gamma C}{N}\sum_{i=1}^{N}|\nu_1(s_i',a_i') - \nu_2(s_i',a_i')| + \frac{(1-\gamma)}{N}\sum_{i=1}^{N}|\nu_1(s_0^i,a_i^0) - \nu_2(s_0^i,a_i^0)|,$$

which leads to

$$\mathcal{N}_1\left(\left(L + \frac{2+\gamma-\gamma^2}{1-\gamma}C + (1-\gamma)\right)\epsilon', \mathcal{G}, \{Z_i\}_{i=1}^{N}\right)$$

$$\leq \mathcal{N}_1\left(\epsilon', \mathcal{H}, \{s_i,a_i\}_{i=1}^{N}\right)\mathcal{N}_1\left(\epsilon', \mathcal{F}, \{s_i,a_i\}_{i=1}^{N}\right)\mathcal{N}_1\left(\epsilon', \mathcal{F}, \{s_i',a_i'\}_{i=1}^{N}\right)\mathcal{N}_1\left(\epsilon', \mathcal{F}, \{s_0^i,a_0^i\}_{i=1}^{N}\right). \quad (31)$$

Applying lemma 5, we can bound the covering number. Denote the pseudo-dimension of $\mathcal{F}$ and $\mathcal{H}$ as $D_\nu$ and $D_\zeta$, then, we have

$$\mathcal{N}_1\left(\left(L + \frac{2+\gamma-\gamma^2}{1-\gamma}C + (1-\gamma)\right)\epsilon', \mathcal{G}, \{Z_i\}_{i=1}^{N}\right) \leq e^4(D_\mathcal{F}+1)^3(D_\mathcal{H}+1)\left(\frac{4eM_1}{\epsilon'}\right)^{3D_\mathcal{F}+D_\mathcal{H}},$$

which implies

$$\mathcal{N}_1\left(\frac{\epsilon}{8}, \mathcal{G}, \{Z_i\}_{i=1}^{N}\right)$$

$$\leq e^4(D_\mathcal{F}+1)^3(D_\mathcal{H}+1)\left(\frac{32\left(L + \frac{2+\gamma-\gamma^2}{1-\gamma}C + (1-\gamma)\right)eM_1}{\epsilon}\right)^{3D_\mathcal{F}+D_\mathcal{H}} := C_1\left(\frac{1}{\epsilon}\right)^{D_1}, \quad (32)$$

where $C_1 = e^4(D_\mathcal{F}+1)^3(D_\mathcal{H}+1)\left(32\left(L + \frac{2+\gamma-\gamma^2}{1-\gamma}C + (1-\gamma)\right)eM_1\right)^{D_1}$ and $D_1 = 3D_\mathcal{F} + D_\mathcal{H}$.

Combine this result with equation 30, we immediately obtain the statistical error, *i.e.*,

$$\mathbb{P}\left(\sup_{\nu \in \mathcal{F}, \zeta \in \mathcal{H}}\left|\hat{J}(\nu,\zeta) - J(\nu,\zeta)\right| \geq \epsilon\right) \leq 8C_1\left(\frac{1}{\epsilon}\right)^{D_1}\exp\left(\frac{-N\epsilon^2}{512M_1^2}\right).$$

By setting $\epsilon = \sqrt{\frac{C_2\left(\log N + \log\frac{1}{\delta}\right)}{N}}$ with $C_2 = \max\left((8C_1)^{\frac{2}{D_1}}, 512M_1D_1, 512M_1, 1\right)$, we have

$$8C_1\left(\frac{1}{\epsilon}\right)^{D_1}\exp\left(\frac{-N\epsilon^2}{512M_1^2}\right) \leq \delta.$$

$\square$

**Bounding $\epsilon_{stat}$.** As $\hat{\nu}^*$ is a random variable, we need to bound the following instead:

$$\sqrt{\epsilon_{stat}}$$

$$= \left|\hat{\mathbb{E}}_{s,a,s',a'}\left[(\hat{\nu}^*(s,a) - \gamma\hat{\nu}^*(s',a'))r(s,a)\right] - \mathbb{E}_{s,a,s',a'}\left[(\hat{\nu}^*(s,a) - \gamma\hat{\nu}^*(s',a'))r(s,a)\right]\right|$$

$$\leq \sup_{\nu \in \mathcal{F}}\left|\hat{\mathbb{E}}_{s,a,s',a'}\left[(\nu(s,a) - \gamma\nu(s',a'))r(s,a)\right] - \mathbb{E}_{s,a,s',a'}\left[(\nu(s,a) - \gamma\nu(s',a'))r(s,a)\right]\right|,$$

which can be done using a similar argument as above.

**Lemma 7** (Statistical error $\epsilon_{stat}$). *Under Assumption 1, with at least probability $1 - \delta$,*

$$\epsilon_{stat} = \mathcal{O}\left(\frac{\log N + \log \frac{1}{\delta}}{N}\right).$$

*Proof.* We first show that $\forall \nu \in \mathcal{H}$, $(\nu(s,a) - \gamma\nu(s',a')) r(s,a)$ is bounded by $M_2 = \frac{1+\gamma}{1-\gamma}C^2$, *i.e.*,

$$\|(\nu - \gamma\nu') \cdot r\|_\infty \leq (1 + \gamma) C \|\nu\|_\infty \leq \frac{1+\gamma}{1-\gamma}C^2.$$

Then, we apply the lemma 4 with $\mathcal{Z} = S \times A \times S \times A$, $Z_i = (s_i, a_i, s'_i, a'_i)$, and $\mathcal{G} = (\nu - \gamma\nu) \cdot r$,

$$\mathbb{P}\left(\sup_{\nu \in \mathcal{F}}\left|\hat{\mathbb{E}}_Z\left[\left(\nu - \hat{\mathcal{B}}^\pi\nu\right) \cdot r\right] - \mathbb{E}\left[(\nu - \mathcal{B}^\pi\nu) \cdot r\right]\right| \geq \epsilon\right) \tag{33}$$

$$\leq \quad 8\mathbb{E}\left[\mathcal{N}_1\left(\frac{\epsilon}{8}, \mathcal{G}, \{Z_i\}_{i=1}^N\right)\right]\exp\left(\frac{-N\epsilon^2}{512M_2^2}\right). \tag{34}$$

Similarly, we have

$$\frac{1}{N}\sum_{i=1}^N |(\nu_1 - \gamma\nu_1) \cdot r(Z_i) - (\nu_2 - \gamma\nu_2) \cdot r(Z_i)|$$

$$\leq \quad \frac{C}{N}\sum_{i=1}^N |\nu_1(s_i, a_i) - \nu_2(s_i, a_i)| + \frac{\gamma C}{N}|\nu_1(s'_i, a'_i) - \nu_2(s'_i, a'_i)|,$$

leading to

$$\mathcal{N}_1\left((1+\gamma)C\epsilon', \mathcal{G}, \{Z_i\}_{i=1}^N\right) \leq \mathcal{N}_1\left(\epsilon', \mathcal{F}, \{s_i, a_i\}_{i=1}^N\right)\mathcal{N}_1\left(\epsilon', \mathcal{F}, \{s'_i, a'_i\}_{i=1}^N\right). \tag{35}$$

Applying lemma 5, we bound the covering number as

$$\mathcal{N}_1\left((1+\gamma)C\epsilon', \mathcal{G}, \{Z_i\}_{i=1}^N\right) \leq e^2 (D_\mathcal{F} + 1)^2 \left(\frac{2eM_2}{\epsilon'}\right)^{2D_\mathcal{F}}, \tag{36}$$

which implies

$$\mathcal{N}_1\left(\frac{\epsilon}{8}, \mathcal{G}, \{Z_i\}_{i=1}^N\right) \leq e^2 (D_\mathcal{F} + 1)^2 \left(\frac{16(1+\gamma)CeM_2}{\epsilon}\right)^{2D_\mathcal{F}} := C_3\left(\frac{1}{\epsilon}\right)^{D_2},$$

with $C_3 := e^2 (D_\mathcal{F} + 1)^2 (16(1+\gamma)CeM_2)^{D_2}$ and $D_2 = 2D_\mathcal{F}$.

We achieve the statistical error bound, *i.e.*,

$$\mathbb{P}(\sqrt{\epsilon_{stat}} \geq \epsilon) \leq 8C_3\left(\frac{1}{\epsilon}\right)^{D_2}\exp\left(\frac{-N\epsilon^2}{512M_2^2}\right). \tag{37}$$

By setting $\epsilon = \sqrt{\frac{C_4\left(\log N + \log\frac{1}{\delta}\right)}{N}}$ with $C_4 = \max\left((8C_3)^{\frac{2}{D_2}}, 512M_2D_2, 512M_2, 1\right)$, we have

$$8C_3\left(\frac{1}{\epsilon}\right)^{D_2}\exp\left(\frac{-N\epsilon^2}{512M_2^2}\right) \leq \delta.$$

Therefore, we have $\epsilon_{stat} = \mathcal{O}\left(\frac{\log N + \log\frac{1}{\delta}}{N}\right)$, with $1 - \delta$ probability. $\qquad\square$

### C.3. Putting It All Together

**Theorem 2** *Under Assumptions 1 and 3, with $f(x) = \frac{1}{2}x^2$, the mean squared error of DualDICE's estimate is bounded by*

$$\mathbb{E}\left[\left(\hat{\mathbb{E}}_{d^{\mathcal{D}}}\left[\hat{\zeta}(s,a) \cdot r\right] - \rho(\pi)\right)^2\right] = \widetilde{\mathcal{O}}\left(\epsilon_{approx}(\mathcal{F}, \mathcal{H}) + \epsilon_{opt} + \frac{1}{\sqrt{N}}\right),$$

*where $\mathbb{E}[\cdot]$ is taken w.r.t. randomness both in the sampling of $\mathcal{D} \sim d^{\mathcal{D}}$ and in the algorithm, $\widetilde{\mathcal{O}}(\cdot)$ ignores logarithmic factors, and the error terms are defined in equation 40.*

*Proof.* By equations 25 and 28, the error can be decomposed as

$$\mathbb{E}\left[\left(\hat{\mathbb{E}}_{d^{\mathcal{D}}}\left[\hat{\zeta}(s,a) \cdot r\right] - \mathbb{E}_{d^{\mathcal{D}}}\left[w_{\pi/\mathcal{D}}(s,a) \cdot \overline{R}(s,a)\right]\right)^2\right]$$

$$\leq 2\mathbb{E}\left[\left(\hat{\mathbb{E}}_{d^{\mathcal{D}}}\left[\hat{\zeta}(s,a) \cdot r\right] - \hat{\mathbb{E}}_{d^{\mathcal{D}}}\left[\left(\hat{\nu} - \hat{\mathcal{B}}^\pi \hat{\nu}\right)(s,a) \cdot r\right]\right)^2\right]$$

$$+ 2\mathbb{E}\left[\left(\hat{\mathbb{E}}_{d^{\mathcal{D}}}\left[\left(\hat{\nu} - \hat{\mathcal{B}}^\pi \hat{\nu}\right)(s,a) \cdot r\right] - \mathbb{E}_{d^{\mathcal{D}}}\left[w_{\pi/\mathcal{D}}(s,a) \cdot \overline{R}(s,a)\right]\right)^2\right]$$

$$\leq 8C_r^2 \mathbb{E}\left[\left(\left\|\hat{\zeta} - \hat{\zeta}^*\right\|_{\hat{\mathcal{D}}}^2 + \left\|\left(\hat{\nu}^* - \hat{\mathcal{B}}^\pi \hat{\nu}^*\right) - \left(\hat{\nu} - \hat{\mathcal{B}}^\pi \hat{\nu}\right)\right\|_{\hat{\mathcal{D}}}^2\right)\right] + 8C_r^2 \mathbb{E}\left[\left\|\hat{\zeta}^* - \left(\hat{\nu}^* - \hat{\mathcal{B}}^\pi \hat{\nu}^*\right)\right\|_{\hat{\mathcal{D}}}^2\right]$$

$$+ 2\mathbb{E}\left[\left(\hat{\mathbb{E}}_{d^{\mathcal{D}}}\left[\left(\hat{\nu} - \hat{\mathcal{B}}^\pi \hat{\nu}\right)(s,a) \cdot \hat{r}(s,a)\right] - \mathbb{E}_{d^{\mathcal{D}}}\left[w_{\pi/\mathcal{D}}(s,a) \cdot r(s,a)\right]\right)^2\right]. \tag{38}$$

We can bound the last term, $\mathbb{E}\left[\left(\hat{\mathbb{E}}_{d^{\mathcal{D}}}\left[\left(\hat{\nu} - \hat{\mathcal{B}}^\pi \hat{\nu}\right)(s,a) \cdot \hat{r}(s,a)\right] - \mathbb{E}_{d^{\mathcal{D}}}\left[w_{\pi/\mathcal{D}}(s,a) \cdot r(s,a)\right]\right)^2\right]$, by straightforwardly combining equation 29, lemma 6 and lemma 7 into equation 24. Specifically, by lemma 6, we have

$$\mathbb{E}\left[\epsilon_{est}(\mathcal{F})\right] = \sqrt{\frac{C_2 \log N + \log \frac{1}{\delta_1}}{N}}(1 - \delta_1) + 2\delta_1 M_1 = \mathcal{O}\left(\sqrt{\frac{\log N}{N}}\right),$$

by setting $\delta_1 = \frac{1}{\sqrt{N}}$. Similarly, we have

$$\mathbb{E}\left[\epsilon_{stat}\right] = \frac{C_4\left(\log N + \log \frac{1}{\delta_2}\right)}{N}(1 - \delta_2) + 2\delta_2 M_2 = \mathcal{O}\left(\frac{\log N}{N}\right),$$

where the last equation comes from by setting $\delta_2 = \frac{1}{N}$. Plug these results into equation 24, we have

$$\mathbb{E}\left[\left(\hat{\mathbb{E}}_{d^{\mathcal{D}}}\left[\left(\hat{\nu} - \hat{\mathcal{B}}^\pi \hat{\nu}\right)(s,a) \cdot \hat{r}(s,a)\right] - \mathbb{E}_{d^{\mathcal{D}}}\left[w_{\pi/\mathcal{D}}(s,a) \cdot r(s,a)\right]\right)^2\right]$$

$$\leq \mathcal{O}\left(\epsilon_{approx}(\mathcal{F}) + \epsilon_{approx}(\mathcal{H}) + \epsilon_{opt}\right) + \widetilde{\mathcal{O}}\left(\frac{1}{N} + \sqrt{\frac{1}{N}}\right), \tag{39}$$

where $\epsilon_{opt} = \mathbb{E}[\hat{\epsilon}_{opt}]$, $\epsilon_{approx}(\mathcal{F}) := \sup_{\nu \in S \times A \to \mathbb{R}} \inf_{\nu \in \mathcal{F}}\left(\|\nu_{\mathcal{F}} - \nu\|_{\mathcal{D},1} + \|\nu_{\mathcal{F}} - \nu\|_{\beta\pi,1}\right)$, and $\epsilon_{approx}(\mathcal{H}) := \sup_{\zeta \in S \times A \to \mathbb{R}} \inf_{\zeta \in \mathcal{H}}\left(\|\zeta_{\mathcal{H}} - \zeta\|_{\mathcal{D},1} + \|\zeta_{\mathcal{H}} - \zeta\|_{\beta\pi,1}\right)$, due to the approximation with $\mathcal{F}$ for $\nu$ and $\mathcal{H}$ for $\zeta$, respectively.

The first term in equation 38, $\mathbb{E}\left[\left(\left\|\hat{\zeta} - \hat{\zeta}^*\right\|_{\hat{\mathcal{D}}}^2 + 8\left\|\left(\hat{\nu}^* - \hat{\mathcal{B}}^\pi \hat{\nu}^*\right) - \left(\hat{\nu} - \hat{\mathcal{B}}^\pi \hat{\nu}\right)\right\|_{\hat{\mathcal{D}}}^2\right)\right]$, is also the optimization error $\epsilon_{opt}$.

The second term, $\mathbb{E}\left[\left\|\hat{\zeta}^* - \left(\hat{\nu}^* - \hat{\mathcal{B}}^\pi \hat{\nu}^*\right)\right\|_{\hat{\mathcal{D}}}^2\right]$, is due to the parametrization by $\mathcal{F}$ and $\mathcal{H}$.

Define the approximation error

$$\epsilon_{approx}\left(\mathcal{F},\mathcal{H}\right) = \epsilon_{approx}\left(\mathcal{F}\right) + \epsilon_{approx}\left(\mathcal{H}\right) + \mathbb{E}\left[\left\|\hat{\zeta}^{*} - \left(\hat{\nu}^{*} - \hat{\mathcal{B}}^{\pi}\hat{\nu}^{*}\right)\right\|_{\hat{\mathcal{D}}}^{2}\right], \tag{40}$$

combine equation 39 and the extra errors, we immediately have

$$\mathbb{E}\left[\left(\hat{\mathbb{E}}_{d^{\mathcal{D}}}\left[\hat{\zeta}\left(s,a\right)\cdot\hat{r}\left(s,a\right)\right] - \mathbb{E}_{d^{\mathcal{D}}}\left[w_{\pi/\mathcal{D}}(s,a)\cdot r(s,a)\right]\right)^{2}\right] = \widetilde{\mathcal{O}}\left(\epsilon_{approx}\left(\mathcal{F},\mathcal{H}\right) + \epsilon_{opt} + \frac{1}{\sqrt{N}}\right),$$

which is the first conclusion.

$\square$

### C.4. Optimization Error

In this section, we characterize the optimization error $\hat{\epsilon}_{opt}$. With different parametrizations for $(\mathcal{F},\mathcal{H})$ and different optimization algorithms for $\hat{J}\left(\nu,\zeta\right)$, the convergence rate of $\epsilon_{1}$ will be different. For general parametrization of $(\mathcal{F},\mathcal{H})$ as neural network, how to quantitively analyze the optimization error is still an open problem and out of the scope of this paper. We focus on the tabular, linear or kernel parametrization for $(\mathcal{F},\mathcal{H})$. Let $(\mathcal{F},\mathcal{H})$ are the family of linear models with basis function $\psi\left(s,a\right) \in \mathbb{R}^{p}$. The tabular and kernel version can be easily generalized by treating $\psi$ as indicator vectors or infinite dimension feature mapping, respectively, and we omit here. Then, we can parametrize $\nu\left(s,a\right) = w_{\nu}^{\top}\psi\left(s,a\right)$ and $\zeta\left(s,a\right) = w_{\zeta}^{\top}\psi\left(s,a\right)$ with $w_{\nu}, w_{\zeta} \in \mathbb{R}^{p}$. Then, the optimization reduces to

$$\min_{w_{\nu}\in\mathcal{F}}\max_{w_{\zeta}\in\mathcal{H}}\hat{J}\left(w_{\nu},w_{\zeta}\right) := w_{\nu}^{\top}\mathcal{A}w_{\zeta} - \frac{1}{N}\sum_{i=1}^{N}f^{*}\left(w_{\zeta}^{\top}\psi\left(s_{i},a_{i}\right)_{\mathcal{H}}\right) - w_{\nu}^{\top}b, \tag{41}$$

where $\mathcal{A} = \frac{1}{N}\sum_{i=1}^{N}\left(\psi\left(s_{i},a_{i}\right) - \gamma\psi\left(s_{i}',a_{i}'\right)\right)\psi^{\top}\left(s_{i},a_{i}\right) \in \mathbb{R}^{p\times p}$ and $b = \frac{(1-\gamma)}{N}\sum_{i=1}^{N}\psi\left(s_{0}^{i},a_{0}^{i}\right)$.

We have

$$\begin{aligned}
\hat{\epsilon}_{opt} &= \left\|\hat{\zeta} - \hat{\zeta}^{*}\right\|_{\hat{\mathcal{D}}}^{2} + \left\|\left(\hat{\nu}^{*} - \hat{\mathcal{B}}^{\pi}\hat{\nu}^{*}\right) - \left(\hat{\nu} - \hat{\mathcal{B}}^{\pi}\hat{\nu}\right)\right\|_{\hat{\mathcal{D}}}^{2} \\
&\leq \|\Psi\|_{2}^{2}\left\|\hat{w}_{\zeta} - \hat{w}_{\zeta}^{*}\right\|^{2} + \|\Phi\|_{2}^{2}\left\|\hat{w}_{\nu} - \hat{w}_{\nu}^{*}\right\|^{2} \\
&\leq \max\left(\|\Psi\|_{2}^{2} + \|\Phi\|_{2}^{2}\right)\left(\left\|\hat{w}_{\zeta} - \hat{w}_{\zeta}^{*}\right\|^{2} + \left\|\hat{w}_{\nu} - \hat{w}_{\nu}^{*}\right\|^{2}\right),
\end{aligned} \tag{42}$$

where $\Psi = \left[\psi\left(s_{i},a_{i}\right)\right]_{i=1}^{N} \in \mathbb{R}^{N\times p}$ and $\Phi = \left[\psi\left(s_{i},a_{i}\right) - \gamma\psi\left(s_{i}',a_{i}'\right)\right]_{i=1}^{N} \in \mathbb{R}^{N\times p}$.

In general case, the optimization 41 is convex-concave, therefore, the vanilla stochastic gradient descent converges in rate $\mathcal{O}\left(\frac{1}{\sqrt{T}}\right)$ in terms of the primal-dual gap. Specifically, we have $f\left(x\right) = \frac{1}{2}x^{2}$, which will lead $\frac{1}{N}\sum_{i=1}^{N}f^{*}\left(w_{\zeta}^{\top}\psi\left(s_{i},a_{i}\right)_{\mathcal{H}}\right) = \|w_{\zeta}\|_{\mathcal{C}}^{2}$ with $\mathcal{C} = \frac{1}{N}\sum_{i=1}^{N}\psi\left(s_{i},a_{i}\right)\psi^{\top}\left(s_{i},a_{i}\right) \in \mathbb{R}^{d\times d}$. Under the assumption as (Du et al., 2017),

**Assumption 8.** *$\mathcal{A}$ has full rank, $\mathcal{C}$ is strictly positive definite, and the feature vector $\psi\left(s,a\right)$ is uniformly bounded.*

We discuss the optimization error $\epsilon_{opt} := \mathbb{E}\left[\epsilon_{1}\right]$, where the $\mathbb{E}\left[\cdot\right]$ w.r.t. the randomness in the algorithm, in two algorithms for equation 41,

- **SVRG** We can easily verify that the $T$-step solution of SVRG, $\left(\hat{\nu}_{T},\hat{\zeta}_{T}\right)$, converges to $\left(\hat{\nu}^{*},\hat{\zeta}^{*}\right)$ in linear rate $\mathcal{O}\left(\exp\left(-T\right)\right)$ in terms of $\mathbb{E}\left[\left\|\hat{w}_{\nu}^{T} - \hat{w}_{\nu}^{*}\right\|^{2} + \left\|\hat{w}_{\zeta}^{T} - \hat{w}_{\zeta}^{*}\right\|^{2}\right]$ following (Du et al., 2017), where the expectation w.r.t. the randomness in the SVRG. Specifically, we have

$$\hat{\epsilon}_{opt} = \mathcal{O}\left(\exp\left(-T\right)\right). \tag{43}$$

- **SGD** Although the optimization equation 41 is not strongly convex-concave, we can still prove $\mathcal{O}\left(\frac{1}{T}\right)$ convergence rate.

**Lemma 9.** *Let the stepsize $\tau_t$ decay in $\mathcal{O}\left(\frac{1}{t}\right)$, assume the norm of the stochastic gradient is bounded, under Assumption 8, we have*

$$\hat{\epsilon}_{opt} = \mathcal{O}\left(\frac{1}{T}\right).\tag{44}$$

*Proof.* Denote $\hat{\theta}_t = \left[\hat{w}_\nu^t, \frac{1}{\sqrt{\rho}}\hat{w}_\theta^t\right]$ and

$$G_t = \begin{bmatrix} 1 & 0 \\ 0 & \frac{1}{\sqrt{\rho}} \end{bmatrix} \cdot \underbrace{\begin{bmatrix} 0 & \sqrt{\rho}\hat{\mathcal{A}}_t \\ -\sqrt{\rho}\hat{\mathcal{A}}_t^\top & \rho\hat{\mathcal{C}}_t \end{bmatrix}}_{\hat{Q}} \cdot \begin{bmatrix} w_\nu^t \\ \frac{1}{\sqrt{\rho}}w_\zeta^t \end{bmatrix} - \begin{bmatrix} \hat{b}_t \\ 0 \end{bmatrix}$$

as the unbiased stochastic gradient with $\mathbb{E}\left[G_t\right] = g_t$, we have the update rule as $\theta_{t+1} = \theta_t - \Sigma_t G_t$, $\quad \Sigma_t = \begin{bmatrix} 1 & 0 \\ 0 & \frac{1}{\sqrt{\rho}} \end{bmatrix}\sigma_t$

We denote $\delta_t = \frac{1}{2}\left\|\hat{\theta}_t - \hat{\theta}^*\right\|^2 = \frac{1}{2}\left[\left\|\hat{w}_\nu^t - \hat{w}_\nu^*\right\|^2 + \frac{1}{\rho}\left\|\hat{w}_\zeta^t - \hat{w}_\zeta^*\right\|^2\right]$ and $\Delta_t = \mathbb{E}\left[\delta_t\right]$. Then, we have

$$\delta_{t+1} = \frac{1}{2}\left\|\hat{\theta}_t - \Sigma_t G_t - \hat{\theta}^*\right\|^2 \le \delta_t + \frac{1}{2}\sigma_t^2\left(1 + \frac{1}{\rho}\right)\|G_t\|^2 - \left(\hat{\theta}_t - \hat{\theta}^*\right)^\top(\Sigma_t G_t).$$

Take the expectation on both sides and $\mathbb{E}\left[\|G_t\|^2\right] \le K^2$,

$$\Delta_{t+1} = \Delta_t + \frac{\sigma_t^2}{2}\left(1 + \frac{1}{\rho}\right)K^2 - \mathbb{E}\left[\left(\hat{\theta}_t - \hat{\theta}^*\right)^\top(\Sigma_t g_t)\right].\tag{45}$$

As shown in (Du et al., 2017), under Assumption 8 and set $\rho = \frac{8\lambda_{\max}(\mathcal{A}\mathcal{C}^{-1}\mathcal{A})}{\lambda_{\min}(\mathcal{C})}$, the matrix $Q := \mathbb{E}\left[\hat{Q}\right]$ has positive real eigenvalue and

$$\lambda_{\max}(Q) \le 9\frac{\lambda_{\max}(\mathcal{C})}{\lambda_{\min}(\mathcal{C})}\lambda_{\max}(\mathcal{A}\mathcal{C}^{-1}\mathcal{A}), \quad \lambda_{\min}(Q) \ge \frac{8}{9}\lambda_{\min}(\mathcal{A}\mathcal{C}^{-1}\mathcal{A}).$$

On the other hand, with the first-order optimality condition, we can show that $Q\hat{\theta}^* = \hat{b}$. Then, we have

$$\mathbb{E}\left[\left(\hat{\theta}_t - \hat{\theta}^*\right)^\top(\Sigma_t g_t)\right] = \mathbb{E}\left[\left(\hat{\theta}_t - \hat{\theta}^*\right)^\top \Sigma_t^2\left(Q\hat{\theta}_t - \hat{b}\right)\right]$$

$$= \mathbb{E}\left[\left(\hat{\theta}_t - \hat{\theta}^*\right)^\top \Sigma_t^2 Q\left(\hat{\theta}_t - \hat{\theta}^*\right)\right] \ge 2\lambda_{\min}(Q)\left(1 + \frac{1}{\rho}\right)\sigma_t^2\Delta_t.$$

Plug this into the equation 30, we obtain the recursion,

$$\Delta_{t+1} \le \Delta_t + \frac{\sigma_t^2}{2}\left(1 + \frac{1}{\rho}\right)K^2 - 2\lambda_{\min}(Q)\left(1 + \frac{1}{\rho}\right)\sigma_t^2\Delta_t \le (1 - 2c\sigma_t)\Delta_t + \frac{1 + \frac{1}{\rho}}{2}\sigma_t^2 K^2,\tag{46}$$

with $c = \lambda_{\min}(Q)\left(1 + \frac{1}{\rho}\right)$. By setting $\sigma_t > \frac{1}{2ct}$, $\Delta_T = \mathcal{O}\left(\frac{1}{T}\right)$.

$\square$

Using the above results for linear parametrization, we can reach the following corollary of Theorem 2.

**Corollary 10.** *Under the conditions of Theorem 2 and with linear parametrization of $(\nu, \zeta)$ and under Assumption 8, after $T$-iteration, we have $\hat{\epsilon}_{opt} = \mathcal{O}\left(\exp\left(-T\right)\right)$ for SVRG and $\hat{\epsilon}_{opt} = \mathcal{O}\left(\frac{1}{T}\right)$ for SGD.*