# OpenReview forum: "DualDICE: Efficient Estimation of Off-Policy Stationary Distribution Corrections"
_ICML.cc/2019/Workshop/RL4RealLife — RL4RealLife 2019_

### Official Review · AnonReviewer2 · 2019-05-25
**Interesting new estimator for off policy evaluation in RL**

**Rating:** 5
**Confidence:** 4

**Review:**

The paper presents a new method for estimating the stationary distribution correction terms necessary for correct off-policy evaluation in RL. There is a lot of effort in this paper to make the estimator work together with stochastic approximation techniques which makes the technique practically applicable.

The paper is well written for the most part except for some parts of the experiments which could have been expanded upon in the appendix. Are the duals \nu and \zeta also neural networks that take as input the state and action? Are they tabular?

Pros:
- Deals with an important problem
- Ensures resulting method can work in the stochastic setting
- Good experimental results against strong baseline
Con:
- For a workshop about RL for real life applications, this paper did not try to evaluate on a real application.

---

### Official Review · AnonReviewer1 · 2019-05-27
**Important contribution but experiments could be stronger**

**Rating:** 4
**Confidence:** 3

**Review:**

The paper considers the problem of off-policy policy evaluation where the dataset is not necessarily created by a single known Markov policy. This work derives a method for estimating stationary distributions correction terms in this setting and provides theoretical guarantees for different function classes. Finally, the proposed estimation technique is compared empirically against predecessors.

This paper addresses a key limitation of most existing off-policy policy evaluation methods which require that the dataset is collected using a known policy. As such, the contributions are very significant and the proposed approach provides a principled way to deal with more realistic scenarios of agnostic datasets. The paper is very well written and easy to follow. It provides a clear and intuitive derivation and situated the proposed approach well in existing literature. Overall, I found the paper very insightful.

The main weakness of this paper is the limited empirical comparison. A key motivation of this work is to be able to handle datasets collected by multiple unknown policies in a principled way. However, it seems this is not really tested in the experiments. The experiments only demonstrate that the proposed method performs similar to predecessors who assume access to the behavior policy but has overall more stable predictions in scenarios where the dataset is collected from a single known Markov policy. From a practitioner's point of view,  it would have been interesting to see how the method performs in cases where the behavior policy is not available against the baselines of simply estimating the behavior policy from data and using prior techniques. This of course violates the assumptions of prior methods but is still a natural baseline.

---

### Decision · Program_Chairs · 2019-05-28

Accept